# ADAPTIVE DATA OPTIMIZATION:
# DYNAMIC SAMPLE SELECTION WITH SCALING LAWS

**Yiding Jiang**[†][*] **Allan Zhou**[‡][*]  **Zhili Feng**[†]  **Sadhika Malladi**[§]  **J. Zico Kolter**[†]

Carnegie Mellon University[†]  Stanford University[‡]  Princeton University[§]

`yidingji@cs.cmu.edu, ayz@cs.stanford.edu`

## ABSTRACT

The composition of pretraining data is a key determinant of foundation models' performance, but there is no standard guideline for allocating a limited computational budget across different data sources. Most current approaches either rely on extensive experiments with smaller models or dynamic data adjustments that also require proxy models, both of which significantly increase the workflow complexity and computational overhead. In this paper, we introduce Adaptive Data Optimization (ADO), an algorithm that optimizes data distributions in an online fashion, concurrent with model training. Unlike existing techniques, ADO does not require external knowledge, proxy models, or modifications to the model update. Instead, ADO uses per-domain scaling laws to estimate the learning potential of each domain during training and adjusts the data mixture accordingly, making it more scalable and easier to integrate. Experiments demonstrate that ADO can achieve comparable or better performance than prior methods while maintaining computational efficiency across different computation scales, offering a practical solution for dynamically adjusting data distribution without sacrificing flexibility or increasing costs. Beyond its practical benefits, ADO also provides a new perspective on data collection strategies via scaling laws.

## 1 INTRODUCTION

Foundation models, large neural networks pre-trained on vast amounts of data, are the backbone for a wide range of today's machine learning workload (Bommasani et al., 2021). It is well established that the composition of the pretraining data plays a crucial role in these models' final performance (Gadre et al., 2024); however, there are currently no standard guidelines for selecting good pretraining data. Most existing datasets are filtered and curated snapshots of the web data (Gao et al., 2020; Penedo et al., 2024), often categorized into distinct heterogeneous domains (e.g., CommonCrawl and Github). Even after filtering, a challenge remains: deciding how to allocate computational resources across these different domains. Given the ever increasing cost of pretraining large foundation models (Sharir et al., 2020), optimizing the data composition is a potential avenue for improving performance without increasing computational cost.

There are two popular approaches to adjusting the data distribution. A straightforward approach to this problem is to experiment with smaller models, test different data policies, and then apply the best-performing policy to the larger model (Ye et al., 2024; Ge et al., 2024). However, this strategy has several drawbacks. First, as the number of domains increases, the cost of training on all possible policies can grow linearly or even exponentially (if the level of discretization remains the same), making even small-scale experiments costly. Second, the optimal data policy for a smaller model does not necessarily generalize to larger models (Ye et al., 2024; Albalak et al., 2023). Another approach involves dynamically adjusting the data policy during training based on various statistics (Xie et al., 2024; Chen et al., 2024; Qin et al., 2024; Fan et al., 2023). However, these methods often require training a smaller proxy model to guide the data policy, whose cost is still non-negligible.

A significant limitation of almost all existing methods (with the exception of Albalak et al. (2023)) is their requirement for additional computational steps beyond standard training. This extra complexity entails substantial modifications to the training pipeline, making it challenging to integrate these

---

[*]Equal contribution.

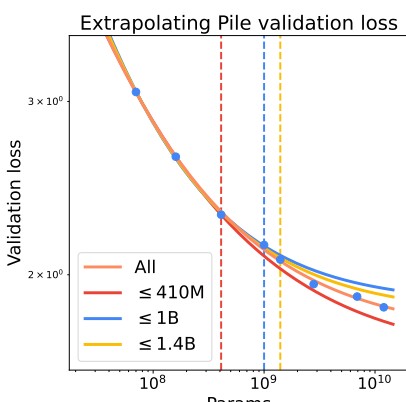

Figure 1: ADO is a cheap online technique for adjusting the data distribution while training large models. In contrast to prior methods, ADO tailors its data distribution to the model as it is being trained, and does not require training smaller proxy models in advance.

methods when other components are varied (e.g., the architecture, tokenizer, or optimizer). The consequence of this usage barrier is that these methods have not been tested widely, making them less popular for those training a new model. To enable broader adoption of data mixture optimization, we argue that it must be implemented in an online fashion, concurrent with model training, without disrupting the standard training process, which means that the data mixture must adapt itself based on the feedback from the model. This is closely related to curriculum learning (Bengio et al., 2009), a training strategy where models are progressively exposed to domains in a curated order.

At a fundamental level, this work develops and empirically investigates questions about the existence of good pretraining curricula and the feasibility of cheaply identifying them. We first demonstrate in controlled experiments that good curricula can generally be found with more computation and that it is hard to accurately predict larger models' performance with only a few small models. This leads to our main contribution, Adaptive Data Optimization (ADO): an algorithm for adaptively adjusting the data mixture online during training (Figure 1). In experiments on language models up to 1.3B parameters trained on the Pile (Gao et al., 2020), we find that ADO improves performance across a variety of common benchmarks and improves validation loss on SlimPajama (Soboleva et al., 2023) and FineWeb (Penedo et al., 2023), both of which are considered to be higher-quality datasets. Most importantly, ADO achieves these without requiring significant additional computation (less than $0.4\%$ wallclock time for 1.3B), proxy models, or extensive modification to the training pipelines.

## 2 MOTIVATIONS

**The drawbacks of small proxy models.** At first glance, transferring data selection strategies from smaller to larger models seems plausible, with positive results in specific cases, such as in Mindermann et al. (2022) and Xia et al. (2024b). However, as shown in Figure 2, using a small number of models to predict the behavior of larger models can exhibit high variance on the Pythia model family (Biderman et al., 2023), indicating that they cannot reliably forecast larger models' performance. At a per-domain level (Figure 7), the variation becomes even more pronounced. While this does not fully rule out the possibility of transfer, it raises doubts about whether small models can reliably select data mixtures for larger models. Another issue is that the efficacy of distributions generated by proxy models is highly sensitive to factors like the tokenizer and other hyperparameters (Albalak et al., 2023). Our findings also confirm this brittleness (Section 4). This means that the optimized distributions from one experiment might not transfer, and they need to be retrained for a new model. This is undesirable because proxy models add non-trivial overhead — Ge et al. (2024) estimated that a single round of DoReMi proxy training requires 760 GPU hours on 8 Nvidia A100 GPUs, and several proxy training rounds are needed before training the full model.

Figure 2: Extrapolating larger models' loss using scaling laws up to a threshold model size, indicated by color. Blue dots correspond to different models. Extrapolations become more accurate with more models.

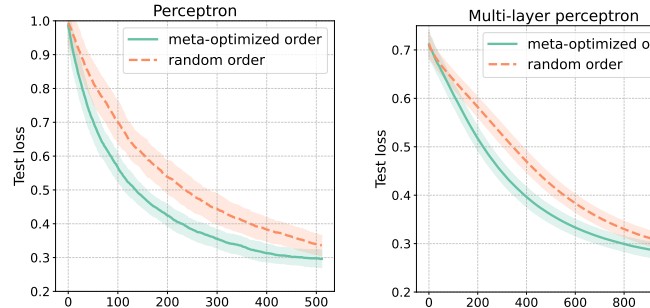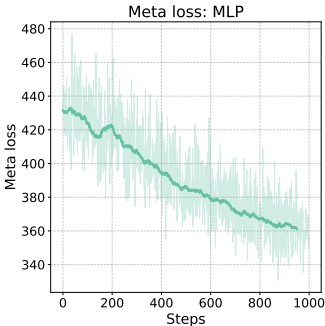

Figure 3: The average learning curves of SGD and the meta-optimized curricula for a logistic regression (**Left**) and an MLP (**Middle**). The meta-optimized curricula consistently outperform SGD. Further, the meta loss for curricula is still decreasing after 1000 steps (**Right**). The first two figures are averaged over 50 runs and the shaded region corresponds to 1 standard deviation.

**The existence of good curricula.** Curriculum learning (Bengio et al., 2009) is a conceptually appealing idea that has been regularly studied over the years, but these methods have not been widely adopted in deep learning outside of a few relatively niche areas (Graves et al., 2017; Florensa et al., 2017; Jesson et al., 2018; Tang et al., 2018). To the best of our knowledge, there is no consensus on why curriculum learning has not had wide success in deep learning. Some hypothesize that deep learning is different from human learning or that overparameterization makes curriculum less effective (Wu et al., 2021; Mannelli et al., 2024). On the other hand, some experimental evidence suggests that deep neural networks implicitly learn functions of increasing complexity from easy to hard (Kalimeris et al., 2019; Baldock et al., 2021), loosely resembling some aspects of human learning (Lawrence, 1952; Baker & Osgood, 1954; Skinner, 1958). For overparameterization, it is not clear if modern foundation model pretraining is in the overparameterized regime since dataset size generally scales with the number of model parameters (Hoffmann et al., 2022).

To motivate our later work, we propose an alternative hypothesis: *good curricula exist but they are computationally difficult to find*. Let's start with an instructive experiment of finding good curricula with more computation. Suppose we have a fixed dataset of size $N$ and want to train SGD at a mini-batch size of 1 for 1 epoch (i.e., each data point is used once, which is common for language model pretraining). We are interested in finding a data ordering that is better than a random shuffle when training a model *initialized at random*. Brute force search is intractable even for a small dataset. Unlike most existing curricula that rely on some prior notions of difficulty, we will find a good curriculum with *only* training data by optimizing over the space of permutations with meta learning. Due to space constraints, we defer the details of this meta learning procedure to Appendix B.

We apply this procedure to a logistic regression problem as well as a multi-layer perceptron where the goal is to imitate a teacher. We randomly sample the initialization $\theta_0$, ground truth $\theta^\star$ and the data $\mathcal{Z} = \{(\boldsymbol{x}_i, f(\boldsymbol{x}_i; \theta^\star))\}_{i=1}^N$. For a fixed set of data points, we run the meta optimization for some steps (200 for logistic regression and 1000 for MLP) and evaluate the performance of learned ordering on a new initialization. We repeat this experiment 50 times and report the results in Figure 3. As can be seen from the results, the performance of meta-optimized data ordering is significantly better than that of random order even though during evaluation the models are initialized at random. In a concurrent work, Gu et al. (2024) conducted a similar experiment but their meta objective uses additional validation data and optimizes a distribution over all training data points rather than SGD.

This shows that given a dataset, it is possible to find an ordering that greatly outperforms random ordering without committing to any prior notion of difficulty. This procedure does *not* leverage additional data, but it requires significantly more computational resources than random ordering, so it is neither practical nor scalable. Moreover, given our limited understanding of the landscape of curriculum optimization, it is highly unlikely that we are identifying the globally optimal curriculum. As shown in Figure 3 (right), even this meta-optimization has not fully converged after 1000 steps. Nonetheless, it demonstrates definitively the existence of a good curriculum, which raises the question of how we can find a better curriculum in a computationally efficient manner[1].

---

[1]Since a static mixture is nothing but a fixed stationary curriculum throughout the training, this computational difficulty applies to finding a static mixture too.

# 3 ADAPTIVE DATA OPTIMIZATION

We demonstrated above a general, albeit costly, strategy for finding good curricula but it is too expensive to be practically useful. We believe that an ideal online data selection strategy for pretraining foundation models should incur a minimal computational cost.

Furthermore, computational efficiency alone is not sufficient. Perhaps an equally important consideration is that such methods should avoid explicit dependence on any particular specification of downstream tasks. The fundamental premise of a foundation model is to serve as the basis for *all* reasonable downstream tasks; explicitly relying on a fixed set of downstream tasks to pick pre-training data distribution risks unintentionally overfitting to the downstream tasks (Goodhart, 1985). Indeed, the evaluation of current language models has become increasingly difficult as performance on existing benchmarks becomes saturated. While evaluating downstream tasks is undoubtedly important because they are representative of the models' main use cases, we believe that it is valuable to define an objective for online data selection that is agnostic to the downstream tasks.

Previous works like DoReMi (Xie et al., 2024), and DoGE (Fan et al., 2023) share this task-agnostic philosophy of data selection. However, these methods require training proxy models which makes them inconvenient and expensive to use. Lifting the requirement for a proxy model would make data selection methods more accessible and practical.

The goal of this work is to advocate for data selection methods that satisfy the following desiderata[2]:

> **Adaptive Data Optimization**
>
> (i) Does not leverage extra information such as additional data or existing models.
>
> (ii) Does not depend on multi-staged training with different proxy models.
>
> (iii) Does not require significant computational resources and can be run online.

Based on these criteria, we introduce our method of online data selection, *Adaptive Data Optimization* (ADO). In the following sections, we present the algorithmic components of ADO. First, we discuss how we fit a scaling law for each domain, allowing the data mixture to account for variations in the natural diversity across different domains. Next, we describe how the data mixture is dynamically adjusted based on these scaling laws *during model training*. Each domain is characterized by two key quantities: 1. the domain's learning potential, indicating the potential improvement from further optimization in that domain (Section 3.1), and 2. a credit assignment score that quantifies the domain's contribution to the reduction of training loss (Section 3.2). To address potential noise from simultaneously updating the data mixture and the model, we employ several time-averaging techniques to smooth these quantities (Section 3.3).

## 3.1 CONSTRUCTION NEURAL SCALING LAWS FOR EACH DOMAIN

To achieve our desiderata, it is crucial to be able to predict the evolution of our model's training online with cheap computational routines that do *not* scale with the size of the model. To accomplish this, we use neural scaling laws (Kaplan et al., 2020) to extrapolate the loss as a function of the amount of training data observed so far, which we denote $n$. It is important to highlight that this scaling law is for predicting the future training loss within a single training run. In contrast, typical neural scaling laws (Hoffmann et al., 2022) use the results of completed training runs to extrapolate the loss of future training runs, typically with larger models or datasets.

For simplicity and interpretability, we use a standard power law: $\widehat{\mathcal{L}}(n; \alpha, \beta, \varepsilon) = \varepsilon + \beta n^{-\alpha}$, where $\alpha, \beta, \varepsilon$ are the parameters to be fitted, and $\widehat{\mathcal{L}}$ is a surrogate loss used to model the behavior of the true loss. The parameter $\varepsilon$ can be seen as the irreducible loss or the "entropy" of the data. Data with high diversity (e.g., CommonCrawl) would naturally have a higher irreducible loss and data that are more predictable (e.g., code and math) would tend to have a lower irreducible loss. The parameter $\alpha$ measures how quickly the loss decreases with more training data. Although such scaling laws may be inaccurate at extrapolating training loss far into the future (e.g., due to the learning rate schedule),

---

[2]These criteria are not mutually exclusive with proxy models or other offline data selection methods and can be used in tandem if needed (e.g., training a model that specializes in coding).

we can cheaply refit the parameters on the fly (Appendix A.1). Since the parameters from each fit are only used for a short period before being updated and the learning curves usually do not change abruptly, our scaling law need only be *locally* accurate.

Instead of fitting a single scaling law for the overall loss (Kaplan et al., 2020; Hoffmann et al., 2022), we will fit a separate scaling law for the loss on each domain, each of which will guide ADO to adjust the weights of each domain.

**Definition 1.** *A **domain scaling law** is a training sample scaling law, $\widehat{\mathcal{L}}_k(n) = \widehat{\mathcal{L}}(n; \alpha_k, \beta_k, \varepsilon_k) = \varepsilon_k + \beta_k n^{-\alpha_k}$, that predicts the $k^{th}$ domain's training loss after training on $n$ samples. $\varepsilon_k, \beta_k, \alpha_k$ are functions of the architecture, training algorithms, and the overall data distribution.*

The domain scaling law serves as a tractable middle ground between modeling the overall loss and modeling how every domain interacts with each other. Modeling the full dependencies faithfully may be challenging while training online because we cannot modify the data distribution drastically. As the first step towards this goal, our method will only model how each domain interacts with itself and leave more complex modeling to future works. Nonetheless, this scaling law reveals important information about each domain. For example, $\varepsilon_k$ corresponds to the estimated irreducible loss of a domain, which could be useful for various applications (Xia et al., 2024a). More importantly, we can interpret the derivative of the loss with respect to the number of samples as the following:

$$\frac{d\widehat{\mathcal{L}}_k(n)}{dn} = \frac{-\alpha_k \beta_k n^{-\alpha_k}}{n} = -\frac{1}{n} \underbrace{\alpha_k}_{\text{Learning speed}} \underbrace{(\widehat{\mathcal{L}}_k(n) - \varepsilon_k)}_{\text{Reducible loss}} . \tag{1}$$

The quantity $\widehat{\mathcal{L}}_k(n) - \varepsilon_k$ informs us about how far away a particular domain is from the estimated minimum loss, which coincides with the notion of population-level *reducible loss* (Mindermann et al., 2022), and $\alpha_k$ indicates how fast the loss of a particular domain is changing with new data. This derivative informs us about how much loss decrease we can expect per data point locally. At a high level, the cross entropy loss has an information theoretic interpretation, so this quantity can be understood as a form of *information density*, or, the amount of information gain per unit of computation. Intuitively, prioritizing data with high information density is a likely better use of computation and could lead to richer representation as it implies more information is "stored in" the model weights. We leave the formalization of this statement to future works. It may be useful to contrast this quantity with Albalak et al. (2023, ODM) which only prioritizes domains with a high loss $\mathcal{L}_k(n)$ without considering the irreducible loss. This objective could lead to the underrepresentation of domains with inherently lower entropy such as code or math[3].

## 3.2 THE CONTRIBUTION OF THE DATA FROM EACH DOMAIN

In contrast to prior works that train multiple models to find optimal data policies offline (Ye et al., 2024; Ge et al., 2024; Liu et al., 2024), our online approach adaptively prioritizes selecting data so that the model learns quickly, as predicted by scaling laws that we fit on the fly. It consists of a data policy $\pi(t) \in \Delta^K$ that specifies a sampling distribution over the $K$ domains. An intuitive approach would be to prioritize sampling domains where the model will learn quickly: e.g., we could sample from domain $k$ in proportion to $-\frac{d\widehat{\mathcal{L}}_k}{dn}$ (Equation 1).

However, independent domain scaling laws do not account for how samples from one domain help learning on a different domain. Consider a thought experiment where the loss $\mathcal{L}_k$ is decreasing rapidly, but we have sampled very little data from domain $k$. Intuitively, data from domain $k$ should not get much credit for this loss decrease, which can instead be attributed to transfer learning from other domains. This calls for some form of credit assignment based on whether a domain has actually been sampled recently.

**Definition 2.** *A **credit assignment score**, $\lambda_k(t)$, is a real positive-valued function that indicates how much data from the $k^{th}$ domain contributed to recent changes in the loss $\mathcal{L}_k$.*

Ideally, we would like to understand how each domain affects every other domain. Unfortunately, this interaction might be highly complex and not local (i.e., the best solution locally at a given time

---

[3]Albalak et al. (2023) showed that ODM achieves higher loss on Github than other methods. In our 124M experiments, ODM achieves 1.54 validation loss on Github while ADO achieves 1.40 validation loss.

---

**Algorithm 1** Adaptive Data Optimization (ADO)

---

1: **Input:** prior $\mu \in \Delta^K$, update interval $t_{\text{update}}$, warmup duration $t_{\text{warmup}}$, $\gamma_1$, $\gamma_2$, $s$, $\delta_{\min}$
2: Initialize $h \leftarrow \mu$, $\bar{\pi} \leftarrow \mu$, $\{\texttt{loss\_k} \leftarrow [\,]\}_{k \in [K]}$
3: Train with $\mu$ for $t_{\text{warmup}}$ steps to initialize each $\texttt{loss\_k}$
4: Initialize domain $k$'s scaling law with $\texttt{loss\_k}$, for $k \in [K]$
5: **for** $t = 0 \rightarrow T$ **do**  ▷ The training loop
6: $\quad$ $\rho(t) \leftarrow$ compute the preference distribution according to Equation 3
7: $\quad$ $\pi(t) \leftarrow \text{clip} \left( \gamma_2\, \rho(t) + (1 - \gamma_2)\bar{\pi}(t-1),\ \delta_{\min} \right)$
8: $\quad$ $\{\ell_k\}_{k \in [K]} \leftarrow$ train the model according to $\pi$ and add domain $k$ loss to $\texttt{loss\_k}$
9: $\quad$ $h(t) \leftarrow \gamma_1 \pi(t) + (1 - \gamma_1)h(t-1)$
10: $\quad$ $\bar{\pi}(t) \leftarrow \frac{1}{t+1}\rho(t) + \left( 1 - \frac{1}{t+1} \right) \bar{\pi}(t-1)$
11: $\quad$ **if** $t \bmod t_{\text{update}} = 0$ **then**
12: $\quad\quad$ Update domain $k$ scaling law with $\texttt{loss\_k}$, for $k \in [K]$

---

step may not be optimal in the long run). As mentioned earlier, we focus on modeling the domain's contribution to itself for now. A reasonable assumption is that the contribution a domain makes to its own learning progress is positively correlated with what proportion of the recent data came from this domain. In other words, we assume that if a domain was useful for itself in recent history, it will likely continue to be useful. Based on this assumption, we keep a history, $h(t)$, of the exponential moving average (EMA) of the *recent* data policy (with a coefficient of $\gamma_1 < 1$) and apply a power transformation with a smoothing parameter $s < 1$ to smooth the distribution since the assignment can be inaccurate:

$$h_k(t) = \gamma_1 \pi_k(t) + (1 - \gamma_1)h_k(t-1), \quad \lambda_k(t) \propto h_k(t)^s. \tag{2}$$

Here a smaller $\gamma_1$ puts more weighting on policy history in the EMA, and decreasing $s$ leads to stronger smoothing of the credit assignment across tasks. We find that $\gamma_1 = 0.1$ and $s = 0.5$ work well consistently for all scales we tested. We emphasize that this design choice is a simple heuristic that can be easily computed and other choices are possible.

## 3.3 CONSTRUCTING THE DATA POLICY UPDATE

To define our data policy, we combine the two principles we have seen thus far: prioritize sampling from a domain $k$ if the model is decreasing its loss $\mathcal{L}_k$ quickly, but only if that decrease can be attributed to data from domain $k$. Additionally, it is common to have a natural prior distribution $\mu \in \Delta^K$ over the domains (e.g., number of tokens in each domain), which we may also incorporate:

**Definition 3.** *Given the learning speed forecast by a scaling law, $\frac{\partial}{\partial n}\widehat{\mathcal{L}}_k(n)$, credit assignment score $\lambda_k(t)$, and a prior domain weight $\mu_k$, the **preference distribution** is*

$$\rho_k(t) \propto -\mu_k \frac{\partial}{\partial n}\widehat{\mathcal{L}}_k(n)\, \lambda_k(t) = \frac{1}{n}\, \mu_k\, \lambda_k(t)\, \alpha_k\, (\widehat{\mathcal{L}}_k(n) - \varepsilon_k). \tag{3}$$

Note that the preference distribution is *not* a perfect indication of the best possible distribution to learn from because we have no access to the true gradient of the data policy. As shown in the definition of the credit assignment score, we have to resort to various approximations.

**Temporal average.** This local estimate of a data policy may not be optimal globally. Both DoReMi (Xie et al., 2024) and DoGE (Fan et al., 2023) found superior performance by averaging the proxy model's training distribution over all time steps, to obtain a stationary distribution for the actual training run. Since we need to select a data policy online, we instead take inspiration from Defazio et al. (2024) and use an online analogy to post-hoc averaging:

$$\bar{\pi}_k(t) = \frac{1}{t+1}\rho_k(t) + \left( 1 - \frac{1}{t+1} \right) \bar{\pi}_k(t-1), \tag{4}$$

$$\pi_k(t) = \gamma_2\, \rho_k(t) + (1 - \gamma_2)\bar{\pi}_k(t-1). \tag{5}$$

Here $\bar{\pi}_k(t)$ is a temporal moving average of the preference policy at every step and $\pi_k(t)$ is a linear combination of the moving average and the chosen policy according to which the data are sampled

Table 1: Zero-shot performance of different methods across different downstream evaluations. Highlighted cells indicate top-performing methods (deeper color indicates better performance).

| | HellaSwag | WinoGrande | PIQA | ARC-E | SciQ | LogiQA2 | LAMBADA | Average |
|---|---|---|---|---|---|---|---|---|
| 124M-Pile | 0.279 | 0.515 | 0.615 | 0.435 | 0.747 | 0.228 | 0.357 | 0.454 |
| 124M-DoReMi | 0.279 | **0.520** | 0.609 | 0.429 | 0.741 | 0.237 | 0.368 | 0.455 |
| 124M-ODM | 0.285 | 0.514 | 0.603 | 0.453 | 0.764 | 0.230 | 0.374 | 0.461 |
| 124M-Balanced | 0.280 | 0.511 | 0.610 | 0.447 | 0.762 | 0.227 | 0.362 | 0.457 |
| 124M-Natural | **0.290** | 0.503 | 0.624 | 0.435 | 0.755 | 0.234 | **0.401** | 0.463 |
| 124M-ADO | **0.290** | **0.520** | **0.635** | **0.456** | **0.771** | **0.244** | 0.371 | **0.470** |
| 1.3B-DoReMi | 0.416 | 0.582 | 0.707 | 0.609 | 0.870 | 0.228 | 0.615 | 0.575 |
| 1.3B-Balanced | 0.382 | 0.546 | 0.689 | 0.580 | 0.862 | 0.225 | 0.613 | 0.557 |
| 1.3B-Natural | 0.424 | 0.584 | 0.718 | **0.627** | **0.886** | **0.232** | 0.624 | 0.585 |
| 1.3B-ADO | **0.442** | **0.590** | **0.730** | 0.625 | 0.875 | 0.228 | **0.638** | **0.590** |

from, which combines both the current preference distribution and the time-averaged distribution. Using the coefficient of $\frac{1}{t}$ ensures that all the preference distributions from all time steps contribute equally to the final average, similar to the effect of post-hoc averaging. Intuitively, a larger value of $\gamma_2$ makes the algorithm more responsive to the current loss dynamics, while a smaller value makes it smoother and less responsive. We find that $\gamma_2 = 0.1$ works well consistently. Additionally, we enforce the smallest domain probability to be no smaller than $\delta_{\min} = 0.01$ to avoid domains being unsampled (Appendix C). Algorithm 1 shows the pseudocode for the entire algorithm.

## 4 EXPERIMENTS

**Evaluation.** We conduct all our experiments on the Pile dataset (Gao et al., 2020) with decoder-only transformer language models (Vaswani et al., 2017) of varying sizes. We consider two types of metrics: 1. validation loss on the Pile, an unweighted version of the pile validation set where each domain receives equal probability, SlimPajama (Soboleva et al., 2023), and a 1 billion token subset of FineWeb (Penedo et al., 2024), 2. zero-shot downstream performance on 6 common-sense reasoning domains from the language model evaluation harness (Gao et al., 2024): HellaSwag (Zellers et al., 2019), WinoGrande (Sakaguchi et al., 2019), PIQA (Bisk et al., 2019), ARC-E (Clark et al., 2018), SciQ (Welbl et al., 2017), LogiQA2 (Liu et al., 2023) and LAMBADA (Paperno et al., 2016).

**Methods.** We conduct experiments at 2 computation scales: 124 million parameters (124M) and 1.3 billion parameters (1.3B). For 124M models, we use a batch size of 256 with a context length of 1024 trained for 60,000 steps (approximately 15 billion tokens), and for 1.3B, we use a batch size of 2048 with a context length of 1024 trained for 60000 steps (approximately 125 billion tokens). Both scales use the natural distribution for $\mu$ and the *same* hyperparameters for ADO. Training details can be found in Appendix A.

**Comparisons.** For comparison, we use the original Pile weights (Gao et al., 2020, Pile) and DoReMi weights (Xie et al., 2024, DoReMi)[4] as the baselines. We also consider ODM (Albalak et al., 2023), another online method. In addition, we introduce a simple baseline that is surprisingly missing from the literature: weighing each domain by the number of tokens in it. This mixture naturally takes into account the tokenizer being used and training on this policy essentially corresponds to training on the natural distribution of the dataset (i.e., corresponds to empirical risk minimization where train, validation, and test sets all follow the same distribution *defined by the tokenizer*). We estimate the tokens per domain by randomly sampling 1000 documents from each domain to estimate each domain's average tokens per document, and then multiply it by the total number of documents (Appendix A). We will refer to this policy as Natural. We also use a second baseline, Balanced, which considers the unweighted mixture of all the domains.

**Observations.** The results for downstream performance are shown in Table 3. We will highlight some key observations here. Observation 1: at 124M scale, ADO achieves the highest average accuracy; it also outperforms all the baselines on 6 out of 7 downstream tasks and is competitive on

---

[4]Since DoReMi is sensitive to the choice of tokenizer, we used the DoReMi weights from Albalak et al. (2023) who use the same tokenizer as we do.

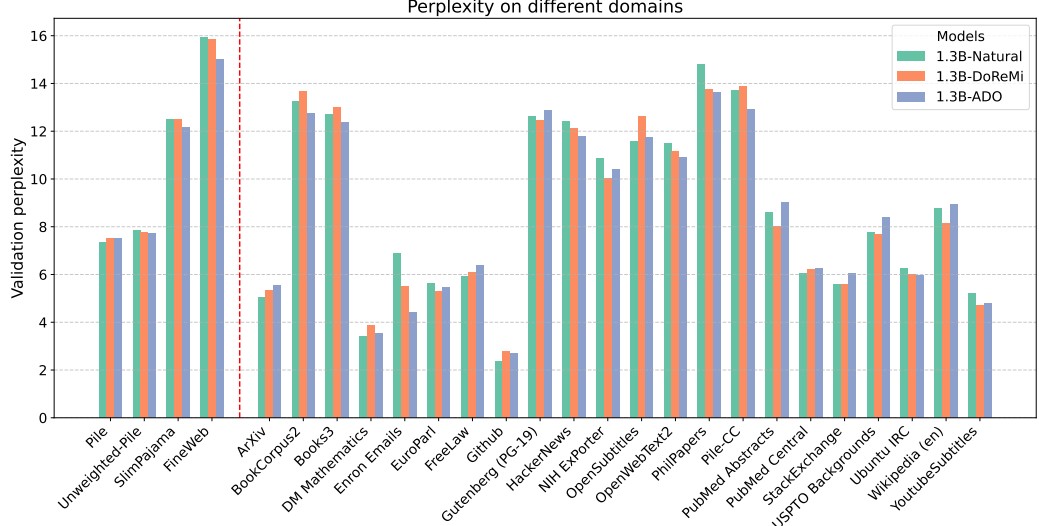

Figure 4: Perplexity of `1.3B` models on different domains, trained on the Pile using either `DoReMi`, `Natural`, or ADO. On the left side of the red line, we have the validation perplexity on the Pile validation set, the unweighted Pile validation set, the SlimPajama validation set, and a random subset of FineWeb. On the right side of the red line, we show the validation perplexity of each domain of the Pile.

LAMBADA. At `1.3B` scale, ADO also achieves the highest average accuracy and outperforms the baselines on 4 out of 7 downstream tasks. Observation 2: interestingly, `Natural` turns out to be a very competitive baseline. It achieves the second-best average downstream performance at both scales and performs well on individual domains, too. To the best of our knowledge, very few works on data selection have benchmarked against using the empirical distribution of tokens given by the dataset and tokenizer. As such, we recommend future works on data selection to compare against this simple baseline. `Balanced` also turns out to be a quite competitive baseline at `124M` scale likely because smaller domains are not extensively repeated given the low total number of tokens processed at this scale. This observation is in line with the findings of Goyal et al. (2024) which suggest that good data policies are different for different compute scales.

Conventionally, validation loss is considered a good indicator of the models' performance if they are trained on the same data, but it is less clear whether it remains a good indicator when the data policy is dynamically changing throughout the training. Empirically (Figure 4), we found that it is difficult to outperform `Natural` on the Pile validation loss (at least within the same training setup and without spending much more computation), even though we do outperform this mixture on the downstream tasks as shown above. Observation 3: for validation loss, ADO slightly underperforms `Natural` on the Pile validation set, but on the validation set of SlimPajama and a subset of FineWeb, ADO outperforms `Natural`. Since SlimPajama and FineWeb are heavily filtered to be "higher quality" data compared to the Pile, we speculate that ADO may be able to implicitly select data aligned with some notion of high quality throughout the training process (e.g., by prioritizing learnability). Nonetheless, since it can be challenging to quantify or precisely define "high quality" data, we will leave this investigation to future work. We include more analysis in Appendix D.1.

## 4.1 ANALYSIS

**Mixture over the training.** A natural question about data selection methods is what does the distribution look like. We now show that the curricula learned by ADO differ for models of different scales in Figure 5. We observe that CommonCrawl (`Pile-CC`) receives the most probability mass in both `124M` and `1.3B`. This observation is consistent with some previous works that put most of the probability mass on CommonCrawl (Xie et al., 2024). `OpenWebText2` also receives a higher proportion compared to the empirical mixture. `Github` receives a higher probability in `124M` initially but eventually decays whereas in `1.3B` it first decreases rapidly and gradually *increases*

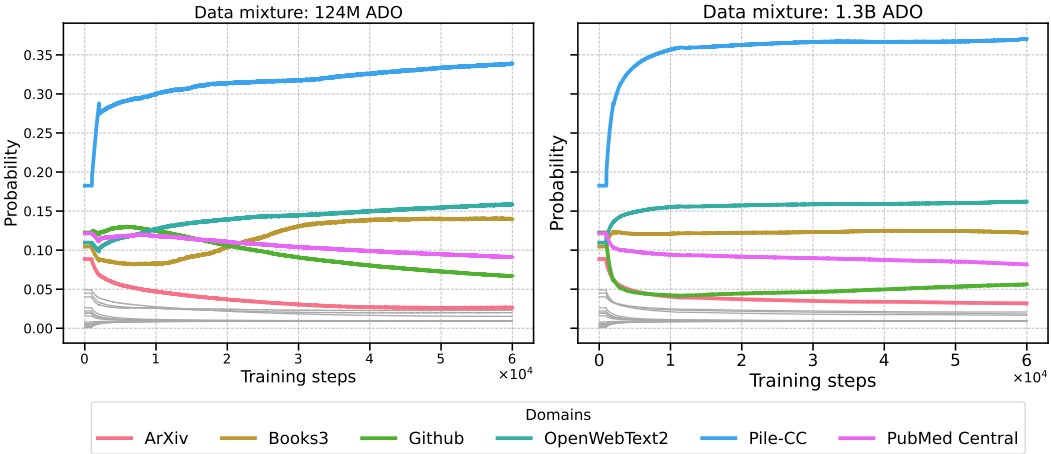

Figure 5: The sampling distribution produced by our data policy during training on The Pile. For ease of visualization, we only highlight the top 6 largest domains. ADO produces qualitatively different strategies at different model scales and adaptively changes its weightings over time.

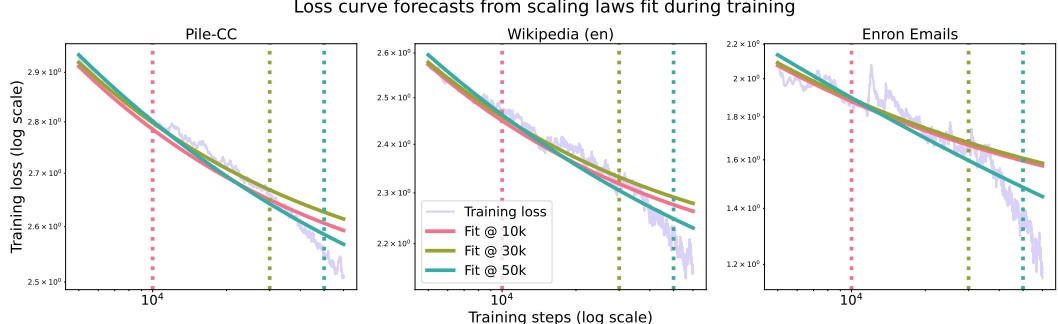

Figure 6: The forecasts from our scaling laws (fit online throughout training) versus the actual training loss on different domains in the Pile. The forecasts shown were fit at 10,000, 30,000, and 50,000 steps (time of fit is shown in dashed lines). We observe that the scaling laws: (1) become more accurate over the course of training, and (2) can be surprisingly accurate at forecasting the final loss even very early into training on some domains, such as Pile-CC. Forecasts from all domains are shown in the Appendix, Figure 8.

towards the end of the training. This is consistent with the intuition that code data naturally have much lower entropy so it is "easier" to learn in some sense. ADO does not account for the fact that code could be desirable for general capabilities such as reasoning (Ma et al., 2024) since it is agnostic to the downstream tasks. This issue can be partially resolved by assigning a higher prior to code but further research on data selection targeting reasoning is likely needed.

**Accuracy of the scaling law throughout the training.** Unlike most applications of scaling laws, we refit the scaling laws on the fly as the models train and the data mixtures are dynamically changing. This introduces new challenges because the forecasts from scaling laws fit early on can be inaccurate for steps far in the future. In Figure 6, we show the predictions of the scaling laws for 3 domains at various points during training (all domains are shown in the Appendix, Figure 8). It can be seen that while the predictions of our scaling laws eventually deviate from the true learning curves, they are accurate locally for much of the training and thus can act as a learning signal for the data policy. More specifically, we can see that the scaling laws consistently *overestimate* the final loss for all domains, likely due to the presence of learning rate schedules. Empirically, when the learning rate decays towards the end of the training, the training loss tends to decrease more rapidly, until a certain point when the learning rate is too small (Hägele et al., 2024). We believe that incorporating recent techniques for incorporating the effect of learning rate schedules into the scaling laws (Tissue et al., 2024) could further improve ADO.

## 5 RELATED WORKS

**Data curation and selection.** For current large language models, compute often poses a greater constraint than data availability, making data selection crucial (Albalak et al., 2024). A widely used approach is *data filtering* (Soboleva et al., 2023; Penedo et al., 2023; 2024; Brandfonbrener et al., 2024), where undesirable data points are removed based on heuristics like perplexity, repetition (Lee et al., 2021), or semantic similarity (Abbas et al., 2023). This filtering process is foundational for constructing most large-scale datasets today. After filtering, data is often categorized into subsets or domains (e.g., code, books), and one must decide how much data to use from each domain.

Two primary strategies for data selection are prevalent: one focuses on deciding whether individual data points should be included based on various criteria (Mindermann et al., 2022; Engstrom et al., 2024), and the other uses all available data but samples from different domains with varying probabilities (Xie et al., 2024; Fan et al., 2023; Albalak et al., 2023). While data selection aims to enhance training efficiency, these methods may introduce considerable computational overhead (Xie et al., 2024; Chen et al., 2024; Fan et al., 2023). Moreover, Kaddour et al. (2023) show that under the same computational budget, these methods often fail to surpass standard training, and Wang et al. (2024) proves that the data selection's efficacy depends on the user's utility function.

Another line of work focuses on selecting pre-training data that aligns more closely with downstream tasks (Kang et al., 2024). Data selection has also been explored for computer vision. For example, Evans et al. (2023) use a small reference model to select data for CLIP, while others propose pruning batches based on diversity criteria (Qin et al., 2024; Hong et al., 2024) to improve training efficiency.

**Neural scaling laws.** Studies have found that various quantities of interest for large pretrained models (e.g., validation loss) can be reliably predicted with simple statistics such as the model size, dataset size, or the amount of computation (Kaplan et al., 2020; Hoffmann et al., 2022). These findings have been central to the design of training protocols for large language models where trial and error are expensive. More recent works have studied the relationship between data repetition and the performance of the models (Hernandez et al., 2022; Muennighoff et al., 2024; Goyal et al., 2024). Data curation, in particular pruning, has also been shown to have significant effects to achieve better scaling laws (Sorscher et al., 2022). Studying the loss curve via scaling law (Hutter, 2021) is relatively less well-explored because the loss curves do not follow a power law exactly due to the learning rate schedule; however, we found that a power law is still a decent model for learning curves with cosine decay for language models. Recently, Tissue et al. (2024) showed that learning rate schedules can be incorporated into scaling laws to make even more accurate predictions, though we do not explore this direction in this work.

## 6 CONCLUSION

Dynamic data selection has the potential to improve the pretraining efficiency of foundation models, but most existing approaches incur additional computation costs. We introduce ADO, a cheap online data selection method that dynamically adjusts data distribution over different domains based on domain scaling laws. The scaling laws forecast the model's loss on different data domains and automatically adjust the training data distribution based on each domain's learning potential. ADO does not require a proxy model so it naturally takes into account the architecture, the optimizer, and other hyperparameters of the training. In our experiments, with a single set of hyperparameters, ADO performs comparably to or better than existing methods on scales from 124M to 1.3B, with much less additional computation. Our implementation only added 20 minutes of additional wall-clock time to a 3.5-day training run ($\sim 0.4\%$), and can be optimized further. The relative cost would decrease more at larger scales since ADO's cost does not scale with the model size. Overall, we believe ADO represents an important step towards accessible and automated online data selection.

**Limitations and future directions.** Given further computational resources, it would be interesting to scale ADO up to larger models (such as 8B models in DoReMi (Xie et al., 2024)) and datasets and study its behaviors more thoroughly. On the technical side, we believe there are several promising directions: 1. how to create better and more fine-grained domains, 2. how to design more realistic scaling laws that can model inter-domain interactions and learning rate schedules, 3. whether insights of ADO can be applied to other training settings such as continued pretraining or fine-tuning, and 4. why do these domain scaling laws arise from naturally occurring data distributions.

ACKNOWLEDGMENTS

We are grateful to Jack Rae, Stephen McAleer, Zhangir Azerbayev, Minqi Jiang, Roberta Raileanu, Michael Xie, Andrew Jesson, Alex Robey, Sam Sokota, Swaminathan Gurumurthy, Jeremy Cohen, and Marc Finzi for helpful discussions during the early phase of this project. We would also like to thank Mengzhou Xia and Tianyu Gao for discussion on model evaluation. The assets used in Figure 1 are courtesy of `pojok d`, `Muhammad Ali`, `faisalovers` from `flaticon.com`. YJ is supported by the Google PhD Fellowship. AZ would like to acknowledge support from Chelsea Finn and the National Science Foundation (GRFP). SM acknowledges funding from NSF, ONR, Simons Foundation, and DARPA. We also thank the Google TPU Research Cloud for generously providing computing for the experiments in this paper.

REPRODUCIBILITY

We have included the main code of the ADO algorithm in the supplementary material alongside this submission, which includes the code for tracking per-domain losses, fitting scaling laws, and calculating the current data policy. This is the only non-standard component of our training pipeline. Our experiments utilize and modify existing open-source training pipelines with details such as hyperparameters provided in Appendix A. The code for this work is available at `https://github.com/yidingjiang/ado`.

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

## A    TRAINING DETAILS

**Architecture**. All experiments use the same architecture, which follows the Llama 2 family (Touvron et al., 2023): a Transformer-based language model with key components including SwiGLU MLP layers (Shazeer, 2020), RMS normalization (Zhang & Sennrich, 2019), and rotary embeddings (Su et al., 2021). We use the GPT-NeoX-20B tokenizer (Black et al., 2022), and tie the weights in the embedding and final layers. Our experiments contain models trained at two scales: `124M` parameters and `1.3B` parameters.

**Training.** All models were trained for 60,000 steps at batch size 2048 (1B) or 256 (124M), using AdamW (Loshchilov, 2017). We use decoupled weight decay with $\lambda = 10^{-4}$, set $\beta_2 = 0.05$, and otherwise use default hyperparameters as specified by Optax (DeepMind et al., 2020). For ADO, we fit scaling laws for each domain every 1,000 training steps starting at step 5,000 (we run an empirical sampling strategy for the first 5,000 steps).

**Compute and time expenditure**. We ran our experiments on TPUs using the open-source midGPT library (Zhou et al., 2023), which is based on JAX (Bradbury et al., 2018) and Equinox (Kidger & Garcia, 2021). All experiments were run on Google Cloud TPUs. On a TPU v3-128, a 1B model can be trained for 60,000 steps in $\sim 3.5$ days. Fitting scaling laws for all domains takes less than 20 seconds–over the course of a training run, this amounts to under 19 minutes in additional time spent fitting scaling laws. This number can likely be further improved by using a smaller or smarter / non-uniform grid (we found the results to be not sensitive to the size of the grid search) or smarter optimization custom-made for power law. Although not present in our particular implementation, the fitting of the scaling laws could, in principle, take place asynchronously in parallel with model training since it does *not* require any information from the model other than the loss, which can be staggered as the scaling laws remain accurate for some duration into the future.

### A.1    FITTING SCALING LAWS

We fit an individual loss curve scaling law to each domain where the input is the total number of samples processed so far, and the prediction is the training loss of that particular domain. Namely, suppose the dataset is broken down into $K$ distinct heterogeneous domains, we would learn $\widehat{\mathcal{L}}_k(n) = \varepsilon_k + \beta_k n^{-\alpha_k}$. To estimate these parameters of the scaling law, we use the standard procedure for fitting a power law (Hoffmann et al., 2022) which minimizes the Huber loss with $\delta = 0.001$ between the predicted log loss and observed log loss with L-BFGS:

$$\min_{\varepsilon_k, \beta_k, \alpha_k} \sum_n \text{Huber}_\delta \left( \log \widehat{\mathcal{L}}_k(n) - \log \mathcal{L}_k(n) \right). \tag{6}$$

Since learning curves have a large number of points, we subset the learning curve at regular intervals to obtain the data for fitting. Similar to prior works, we use grid search with different initialization, although we found that usually a smaller grid search suffices since the individual scaling laws do not need to be very accurate. The losses of the first 500 steps are not included as the weights have not stabilized yet and we subsample the trajectory at 10-step intervals to speed up the training. This does not significantly impact the algorithm since scaling laws do not change drastically. We also apply a threshold to the parameters' values to mitigate potential instability due to online fitting.

$\beta$ and $\varepsilon$ are both parameterized in their log forms. We use a grid search over the following parameters initialization:

- $\alpha_0 \in \{0.1, 0.2, 0.3, 0.4, 0.5, 0.6, 0.7\}$
- $\log \beta_0 \in \{-2, -1, 0, 1, 2, 3, 4, 5\}$
- $\log \varepsilon_0 \in \{-2, -1.5, -1, -0.5, 1, 1.5\}$

Further, we enforce $0 < \alpha < 0.8$, $\log \beta < 6.5$ and $\log \varepsilon > 0.5$. These bounds are purely preventative in cases of numerical instability and are almost never saturated.

The computation complexity of fitting scaling law depends on the rate of convergence of L-BFGS, which can be problem-dependent but is generally efficient and stable. The complexity of fitting per-domain scaling laws scales with the number of domains and the granularity of the initialization search grid. Suppose $G$ is the granularity of discretization for a parameter (e.g., 7 for $\alpha_0$), and K

Table 2: Different mixtures used in this paper.

| Dataset | Natural | DeReMi | Pile |
|---|---|---|---|
| FreeLaw | 0.0449 | 0.0380 | 0.0612 |
| Enron Emails | 0.0010 | 0.0040 | 0.0014 |
| Github | 0.1227 | 0.0325 | 0.0759 |
| OpenSubtitles | 0.0158 | 0.0032 | 0.0155 |
| PubMed Central | 0.1215 | 0.0608 | 0.1440 |
| OpenWebText2 | 0.1096 | 0.1905 | 0.1001 |
| StackExchange | 0.0491 | 0.0746 | 0.0513 |
| Pile-CC | 0.1825 | 0.1379 | 0.1811 |
| ArXiv | 0.0886 | 0.0535 | 0.0896 |
| USPTO Backgrounds | 0.0262 | 0.0327 | 0.0365 |
| Books3 | 0.1046 | 0.0757 | 0.1207 |
| Wikipedia (en) | 0.0402 | 0.1068 | 0.0153 |
| PubMed Abstracts | 0.0221 | 0.0970 | 0.0307 |
| NIH ExPorter | 0.0019 | 0.0084 | 0.0030 |
| BookCorpus2 | 0.0063 | 0.0037 | 0.0075 |
| EuroParl | 0.0081 | 0.0120 | 0.0073 |
| HackerNews | 0.0047 | 0.0084 | 0.0062 |
| DM Mathematics | 0.0191 | 0.0019 | 0.0124 |
| YoutubeSubtitles | 0.0040 | 0.0117 | 0.0060 |
| PhilPapers | 0.0027 | 0.0093 | 0.0038 |
| Ubuntu IRC | 0.0049 | 0.0083 | 0.0088 |
| Gutenberg (PG-19) | 0.0195 | 0.0292 | 0.0217 |

is the number of tasks, the complexity of fitting would be $O(G^3 K)$, which grows linearly with the number of domains. However, we can easily parallelize multiple L-BFGS runs on TPUs (or GPUs), which is why ADO does not add much time in practice. As we mentioned above, 0.04% training time in a 3-day run which is less than 20 minutes, a number that can likely be optimized further.

### A.2 ESTIMATED EMPIRICAL MIXTURE

In Figure 5, we show various data mixtures used in this paper. For the natural distribution, we sampled 1000 documents from each domain, tokenized them with our chosen tokenizer, and then computed the average number of tokens per document. Using the estimated average, we can estimate the total number of tokens within a domain by multiplying it by the number of documents.

Note that the DoReMi differs from the number of Xie et al. (2024) due to the use of different tokenizers. The numbers we use here are from Albalak et al. (2023) which uses GPT-NeoX-20B tokenizer (Black et al., 2022) like us. Also, note that the estimated empirical distribution differs from the default Pile weights. Notably, Github and Wikipedia are weighted significantly higher.

## B    META-LEARNING PERMUTATIONS

Suppose we have a dataset $\mathcal{Z} = \{z_1, z_2, \ldots, z_N\}$ drawn from the distribution $P_z$ and we wish to train a model in the *one-epoch* setting, that is, the model gets to see each example only once[5]. In this setting, the space of curricula is $\mathcal{S}_N$, all permutations of $[N]$. Note that the size of this search space $|\mathcal{S}_N| = N!$ is extremely large for even a small $N$, and would be even larger if we allowed data repetition. Given a curriculum $\sigma : [N] \to [N]$ and the stochastic gradient descent update rule $U(\theta, z) = \theta - \eta \nabla_\theta \ell(z; \theta)$, we initialize the model parameters $\theta_0$ and iteratively update them: $\theta_{t+1} = U(\theta_t, z_{\sigma(t)})$. We can denote the entire training process from $\theta_0$ to $\theta_N$ by $\theta_N = \text{SGD}(\theta_0, \sigma, N)$.

In standard SGD, the curriculum is a randomly sampled permutation, $\sigma_{\text{random}} \sim \text{Unif}(\mathcal{S}_N)$. Given the size of the search space, we can instead formulate the problem of finding curricula

---

[5]This is common for the training regime of the modern language model.

as an optimization problem over $\mathcal{S}_N$. An optimization problem needs an objective, but what should the objective be for finding a good curriculum? The ultimate goal of this problem is to find a curriculum such that the final parameter $\theta_N$ achieves a low population loss, i.e., $\sigma^\star = \arg\min_\sigma \mathbb{E}_{\boldsymbol{z} \sim P_{\boldsymbol{z}}} [\ell(\boldsymbol{z}; \texttt{SGD}(\theta_0, \sigma, N))]$. However, we cannot in general assume access to the data distribution so we need a surrogate objective. In this experiment, we use the following surrogate meta objective that depends only on the training data:

$$\sigma^\star = \arg\min_\sigma \frac{1}{N} \sum_{t=0}^{N-1} \sum_{i=1}^{N} \ell(\boldsymbol{z}_i; \theta_t) = \arg\min_\sigma \mathcal{L}(\sigma), \tag{7}$$

where the dependence of $\theta_t$ on $\sigma$ is implicit since $\theta_{t+1} = U(\theta_t, z_{\sigma(t)})$.

This objective encourages the curriculum to minimize the total training loss after each update (i.e., training as fast as possible). Since we are in the one-epoch setting, the parameters are updated with the gradient of each example only once so there is less, if any, risk of overfitting.

One obstacle to directly learning a good curriculum here is that gradient-based optimizers are not suitable for searching over permutations. We follow Mena et al. (2018) and parameterize the curriculum with a matrix $Z \in \mathbb{R}^{N \times N}$. We then use the Sinkhorn operator $S(\cdot)$ to project $Z$ to a doubly stochastic matrix $X = S(Z)$ (Sinkhorn, 1964; Adams & Zemel, 2011), which we can think of as a "soft" approximation to a permutation. Since $X$ is doubly stochastic but not an actual permutation, we use a weighted batch generalization of our update:

$$\theta_{t+1} = \theta_t - \frac{\eta}{N} \sum_{j=1}^{N} X_{tj} \nabla_\theta \ell(z_j; \theta_t). \tag{8}$$

This is a generalization in the sense that, if $X$ is not only doubly stochastic but also a permutation $\sigma$, then it reduces to SGD with curriculum $\sigma$.

After optimizing $Z$ against the meta objective, we can project $Z$ to a true permutation which serves as our final learned curriculum:

$$Z_{t+1} = Z_t - \eta \nabla_Z \mathcal{L}(S(Z_t)), \quad \sigma = \arg\max_{P \in \mathcal{P}_N} \langle Z_\infty, P \rangle. \tag{9}$$

This procedure can be further generalized to selecting a minibatch. To do so, we relax the constraints on $X$ from a doubly stochastic matrix to a matrix with differentiable top-K relaxation at each row (Xie & Ermon, 2021). We apply this to the MLP experiments in Figure 3 with minibatch size 10.

## C  CLIPPING FUNCTION

Below is the pseudocode we use for clipping the probability. It evenly distributes the excess probability amongst the other categories.

---

**Algorithm 2** Clip Minimum Probability

---

1: **Input:** `probs` $\in \Delta^K$, `min_prob` $\in [0, 1]$
2: **Output:** `clipped_probs`
3: `total_deficit` $\leftarrow \max(\texttt{min\_prob} \cdot n - \sum_k \texttt{probs}, 0)$     ▷ Compute total deficit
4: `scale_factor` $\leftarrow (1 - \texttt{total\_deficit})/(\sum_k \texttt{probs})$     ▷ Compute scale factor
5: `scaled_probs` $\leftarrow \texttt{probs} \cdot \texttt{scale\_factor}$     ▷ Scale the probabilities
6: `clipped_probs` $\leftarrow \max(\texttt{scaled\_probs}, \texttt{min\_prob})$     ▷ Clip probabilities
7: `clipped_probs` $\leftarrow (\texttt{clipped\_probs})/(\sum_k \texttt{clipped\_probs})$     ▷ Normalize
8: **return** `clipped_probs`

---

# D  ADDITIONAL EMPIRICAL RESULTS

## D.1  ADDITIONAL ANALYSIS

In terms of individual tasks, we observe that ADO outperforms ARC-E, SciQ, and LogiQA at 124M scale but not 1.3B scale. The deficit for ARC-E is 0.2% which is relatively small and the performance is far above the other points of comparison. For LogiQA, the performance of DoReMi, Natural, and ADO are all within 0.4% of each other, which is relatively marginal. For SciQ, the difference in performance is more pronounced at 1.1%, but it still outperforms the other baselines. This may suggest that ADO has put less focus on scientific knowledge during pretraining. We believe this could be alleviated by either putting higher prior weights $\mu$ on scientific data such as wikipedia or mixing in high-quality scientific corpus at the later stage of training.

On the other hand, for LAMBADA, ADO outperforms Natural at 1.3B scale by a large margin (1.4%) but not at the 124M scale. This highlights the fact that, for comparing different data mixtures, it is very hard to obtain uniform or monotonic improvement on all downstream tasks the models scale up. Note that downstream tasks are the proxy to the model's capabilities. If there is a particular downstream task of interest, it would be advisable to collect data for the particular task and finetune further. As an evaluation for generic pretraining, aggregated metrics such as average would be a more reliable measurement of the model's performance.

The performance metrics would naturally become more difficult to improve with larger scales because the room to improve becomes smaller. The average error for "natural" at 124M is $1 - 0.463 = 0.537$ and ADO improved over "natural" by 0.07 so the relative improvement is 1.3%. The average error for "natural" at 1.3B is $1 - 0.585 = 0.415$ and ADO improved over "natural" by 0.05 so the relative improvement is 1.2%. This shows that the relative improvement of ADO in average performance at different scales is consistent.

## D.2  ADDITIONAL PLOTS

Table 3: Ablation for the zero-shot performance of different smoothing parameter values, $s$. We observe that 3 out 4 values between 0 and 1 outperform the natural baselines, but larger values of $s$ tend to perform better. This is expected since smaller $s$ is less effective in assigning credit to different domains ($s = 0$ would make $\lambda(t)$ uniform). On the other hand, since the credit assignment is imperfect, using an overly large $s$, could also be undesirable. In our experiments, a value smaller than 1 but not too small achieves the best result.

|         | HellaSwag | WinoGrande | PIQA | ARC-E | SciQ | LogiQA2 | LAMBADA | Average |
|---------|-----------|------------|------|-------|------|---------|---------|---------|
| Natural | 0.2897 | 0.5028 | 0.6235 | 0.4348 | 0.7550 | 0.2341 | **0.4007** | 0.4629 |
| Balanced | 0.2800 | 0.5114 | 0.6099 | 0.4470 | 0.7620 | 0.2271 | 0.3623 | 0.4571 |
| $s = 1.0$ | **0.2951** | 0.5067 | 0.6295 | **0.4747** | 0.7460 | 0.2360 | 0.3635 | 0.4645 |
| $s = 0.5$ | 0.2901 | 0.5201 | **0.6349** | 0.4562 | **0.7710** | **0.2436** | 0.3707 | **0.4695** |
| $s = 0.3$ | 0.2854 | 0.5130 | 0.6311 | 0.4369 | 0.7630 | 0.2277 | 0.3742 | 0.4616 |
| $s = 0.1$ | 0.2894 | **0.5328** | 0.6170 | 0.4457 | 0.7650 | 0.2290 | 0.3625 | 0.4631 |

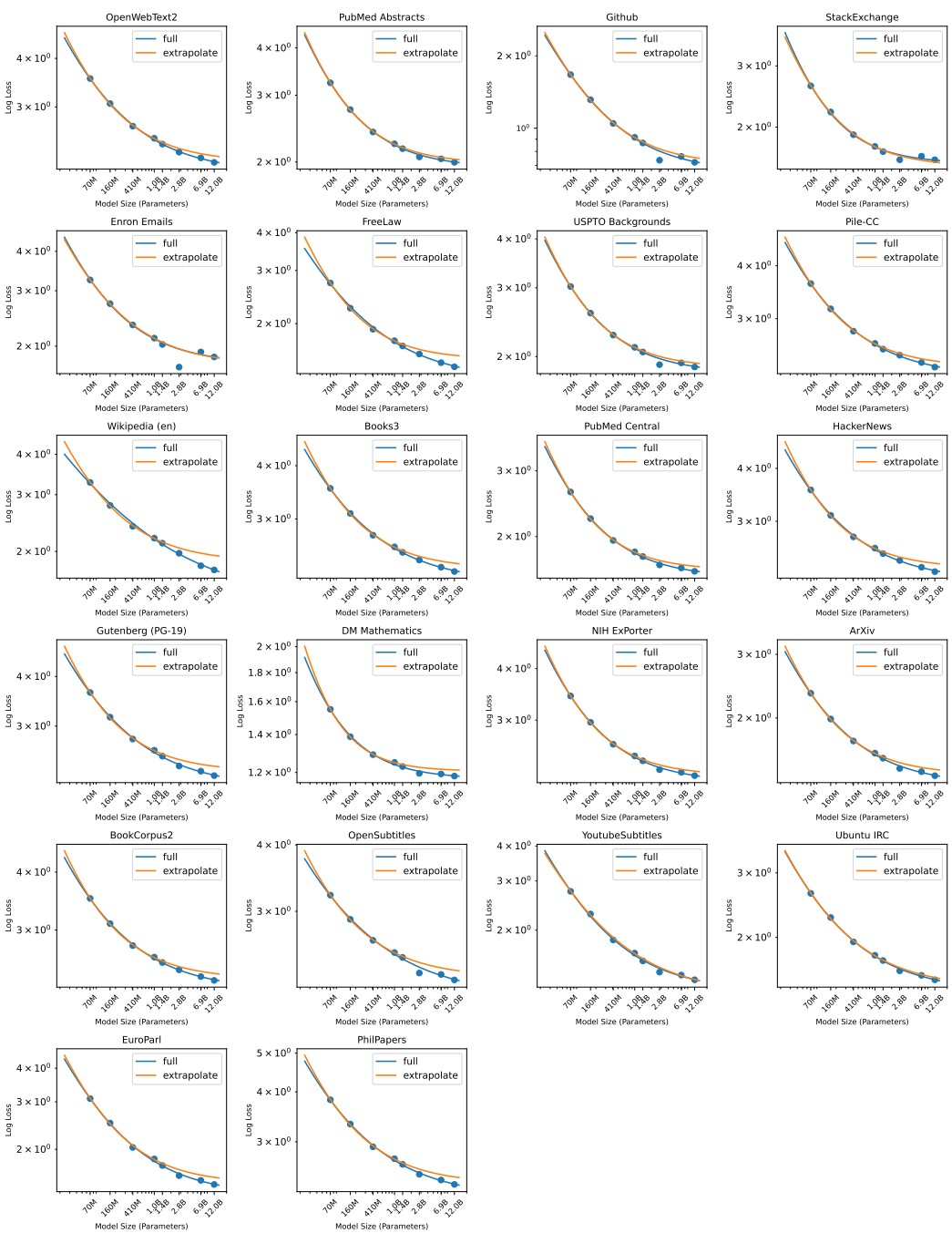

Figure 7: Exrapolating the loss of larger models on each domain of the Pile with the loss of models with fewer than 1B parameters.

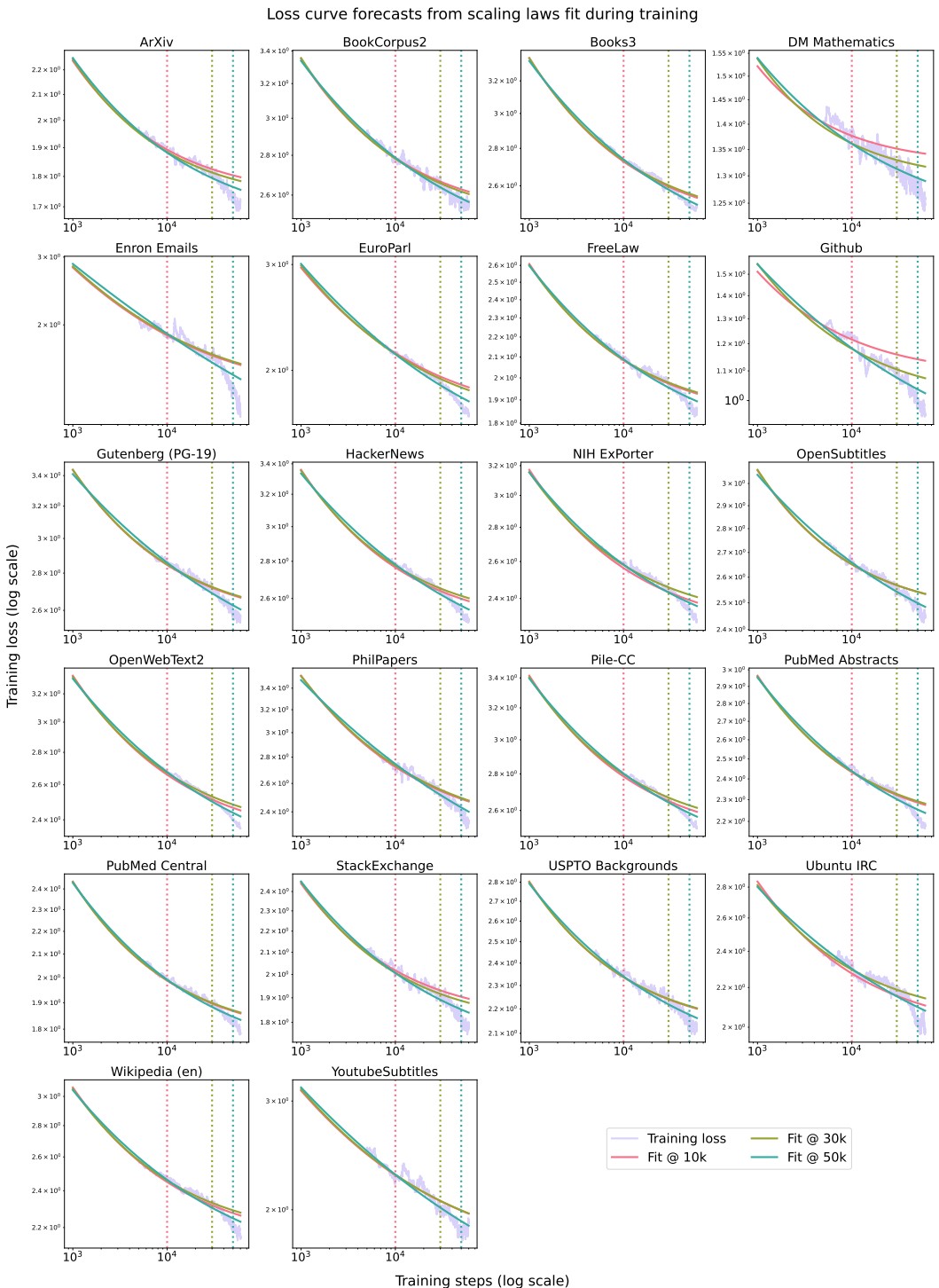

Figure 8: ADO loss forecasts produced by our scaling laws on each domain in The Pile while training a 1B model. Scaling laws are fit regularly throughout training, with the forecasts shown being from fits at 10,000, 30,000, and 50,000 steps (dashed line shows when each scaling law was fit).

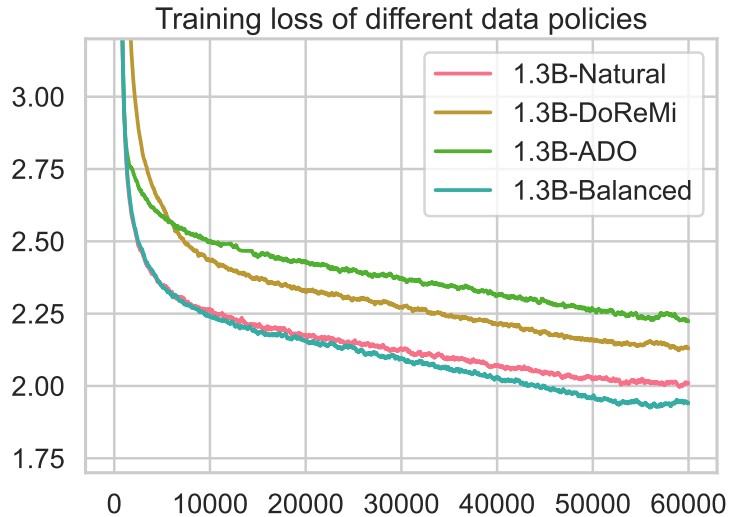

Figure 9: The training losses of different methods differ significantly from each other. ADO has much higher training loss compared to the other methods because ADO selects data with high learning potential, which seem to be correlated with higher entropy.

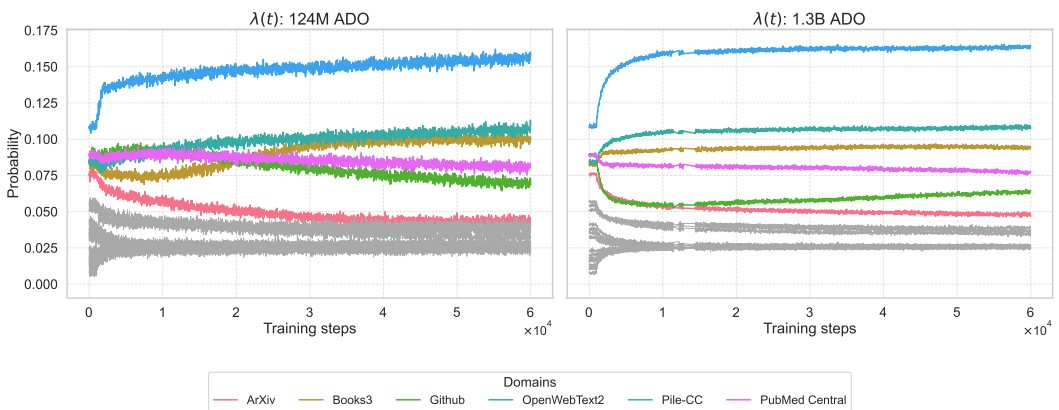

Figure 10: The credit assignment $\lambda(t)$ over the course of training for $s = 0.5$. For clarity, only the top six domains are colored.

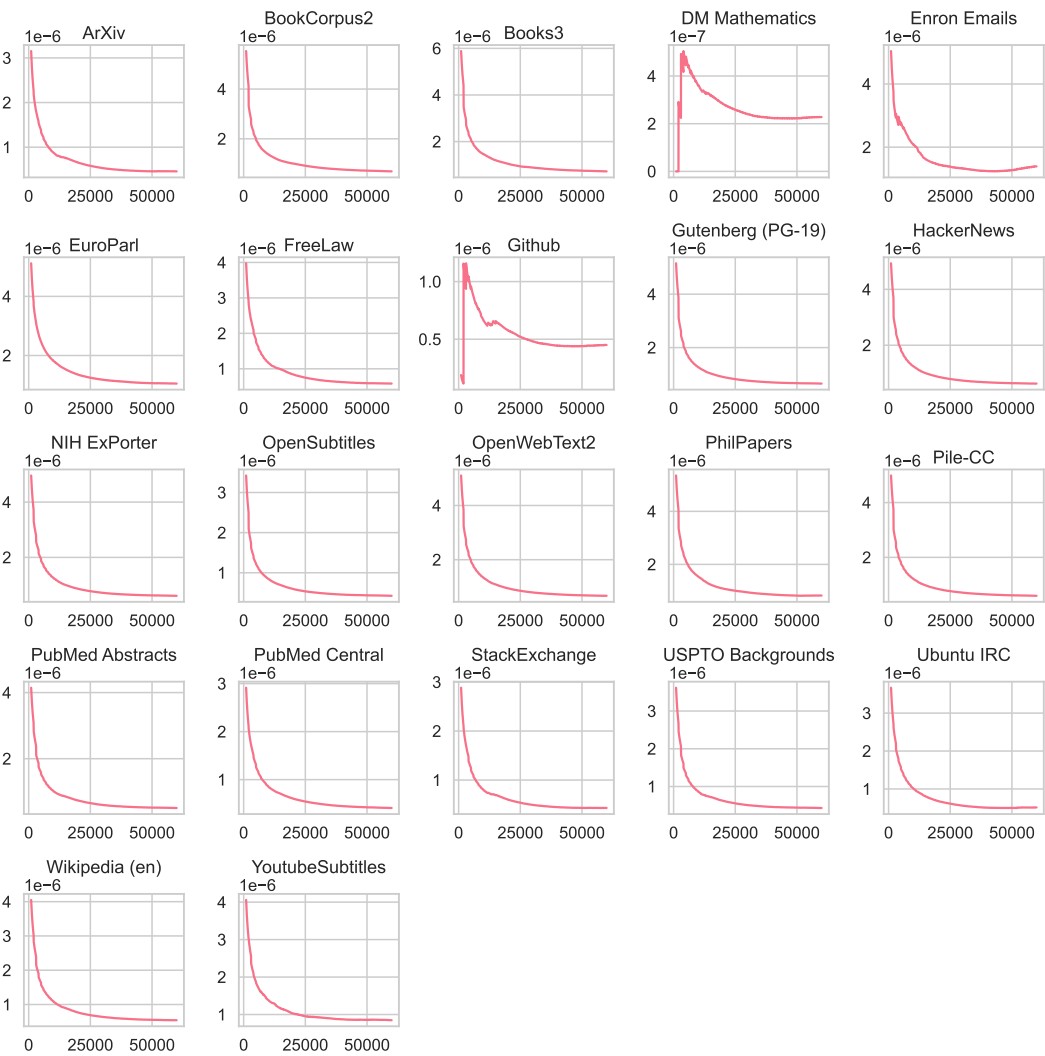

Figure 11: The slope of the scaling law, $\frac{\partial}{\partial n}\widehat{\mathcal{L}}_k(n)$, for each domain throughout 1.3B ADO training.

