# OpenReview forum: "Adaptive Data Optimization: Dynamic Sample Selection with Scaling Laws"
_ICLR.cc/2025/Conference — ICLR 2025 Poster_

### Official Review · Reviewer_Axts · 2024-10-29

**Soundness:** 3
**Presentation:** 4
**Contribution:** 2
**Rating:** 6
**Confidence:** 4

**Summary:**

This work proposes an online adaptive data optimization (ADO) method for finding a policy (weighting) for different domains when pre-training a language model. The core idea is to use scaling law to model training loss. The derivative of the loss w.r.t. to the number of data points can be understood as loss reduction per data point locally, i.e. outline the potential for a domain. The final scoring also considers the contribution of the domain itself from the last update and makes it stable with a moving average. Their experiments show the ADO brings improved downstream performances on 7 common-sense reasoning tasks.

**Strengths:**

- Novelty. It’s a novel idea to improve over online data optimization (ODM, Albalak et al. (2023)) on 3 aspects. 1) Using scaling law to predict training loss on each domain and the derivative of the loss w.r.t. to the number of data points has intuitive interpretation as the loss reduction per data point. 2) The weighting also considers the number of data points used for a domain. 3) A combined weighting with the above two and temporal average.
- Readability. The paper not only shows their method works but also spends a reasonable amount of space discussing perspectives, hence better readability. For example on the curriculum learning (Section 2) and on the requirements (Section 3, online and agnostic to downstream tasks) of data optimization for pretrained models.
- Besides their proposed method, the authors also discover a strong baseline called “natural” policy, that depends on the number of tokens in each domain.

**Weaknesses:**

- Many heuristics are used such as fitting of scaling law (Section 3.1 eq1), credit assignment (Section 3.2, eq2 ), preference distribution (Section 3.3, eq3) and temporal average (eq4 and eq5). The authors try to motivate those choices from related works and intuitions, but only eq1 is adequately explained and validated.
- Following the above, are all the heuristics (eq 1- 5) necessary? How important are they? This is a missing part in the paper. An ablation on them can validate if the invented heuristics are actually all useful.

**Questions:**

- About Figure 2, I think the point on the variance is clear. But what’s more important is the agreement of the relative order of data strategies: if a data strategy is better than another on a smaller model, is it also better on a larger model? Did the authors study that? Also, what are the blue dots? I guess they are the actual validation loss of different model sizes, but would be nice to make it clear in the caption.
- The authors also write “they find $\gamma_1 = 0.1$, $s = 0.5$, $\gamma_2 = 0.1$” works well. How exactly is the process of finding those values?
- The authors write in line 480: “they are accurate locally for much of training and thus can act as a learning signal for the data policy”. On what basis do the claims are made?

---

> ### Author Response · Authors · 2024-11-19
> **Response to Reviewer Axts**
>
> Thank you for the thoughtful response and we are glad that you find our contribution novel and paper well-written. We hope our revision and response will address your concerns.
>
> **W1**: We apologize for any confusion and have updated the text to provide better motivation for these choices. We agree with the reviewer that the usefulness of scaling laws, while heuristic, has been validated by plenty of prior work. Exponential moving averages to smooth noisy online estimates (e.g., Eq 5) are also commonly used, for example in optimizers like Adam. Hence, the novel choices here are our definition of the preference distribution and credit assignment, which boil down to the following intuition given in L301: prioritize sampling from domains where we are making fast progress (learning quickly), but only if the progress can be attributed to the data from that domain. We have added more discussion to the sections to elaborate on our design choices, and would be happy to revise more if you have any further questions.
>
> **W2**: We note that the smoothing we do ($\gamma_1$ and $\gamma_2$) is largely in line with optimization literature and it adds almost no cost. While it is infeasible to exhaustively ablate all hyperparameters given the cost of training, we have conducted a study sweeping over the smoothing parameter $s$ in the appendix (Table 3). We find that multiple values of $s$ work and that even not using any smoothing at all (s=1) still outperforms the baselines, though having some smoothing is even better. Having a small $s$ (e.g., 0.1) corresponds to having almost no credit assignment which does perform as well as having larger $s$, confirming the benefit of credit assignment. Please see our overall response for more details.
>
> **Q1**: The blue dots are indeed the actual validation loss. We have added this to the caption. I think this question is largely an open problem and we unfortunately do not have the resources to do this study. However, anecdotally, the data mixtures used for models of different sizes in the industry labs are significantly different, which would suggest that mixtures that work well for small scales might not work well for larger models. Note that ADO finds different curricula for models of different sizes, which would corroborate this. A concurrent work [Ye et al, 2024] also reaches the same conclusion with empirical verification.
>
> **Q2**: $\gamma_1$ and $\gamma_2$ are fairly standard temporal smoothing parameters used in the literature. We did not have the resources to do an extensive search but the values we picked are standard [Defazio et al, NeurIPS2024] and we did not actually tune them at all. For the smooth parameter $s$, we added a sweep to the revised paper (see Table 3 in the appendix), where we find that multiple values of $s$ work well.
>
> **Q3**: This assessment was made based on comparing the predictions of the scaling laws to the actual observed loss curve (Figure 6). We see that the scaling law predictions are fairly accurate up until near the end of training.
>
> We hope our response, revision, and new ablation address your concerns. We'd also be happy to continue the discussion.
>
> **Reference**
>
> [1] Data Mixing Laws: Optimizing Data Mixtures by Predicting Language Modeling Performance. Ye et al, 2024.
>
> [2] The Road Less Scheduled. Defazio et al, NeurIPS 2024.

---

> > ### Author Response · Authors · 2024-11-24
> > **Follow-up**
> >
> > Thank you again for the thoughtful reviews and valuable feedback. We would appreciate it if you could let us know if our responses and revision have addressed your concerns and whether you still have any other questions about our work.
> >
> > We would be happy to continue the discussion or address any additional comments.

---

> > > ### Comment · Reviewer_Axts · 2024-11-26
> > > **Reply to authors**
> > >
> > > I would like to thank the authors for their response. The added explanations and the reference to previous work help to clarify why those choices in Sections 3.2 and 3.3 are made that way. Hence, I will increase my score to 6.
> > >
> > > However, as also mentioned by the authors "An intuitive approach would be to prioritize sampling domains where the model will learn quickly, ...", how much improvement does the credit assignment score bring is still unknown. I agree with the authors that credit assignment makes sense intuitively, but I think it will be helpful to quantify that in an ablation study.

---

### Official Review · Reviewer_7Mvd · 2024-10-31

**Soundness:** 2
**Presentation:** 3
**Contribution:** 3
**Rating:** 6
**Confidence:** 3

**Summary:**

The paper introduces ADO, a novel online algorithm for optimizing data distributions during the training of large-scale foundation models. ADO addresses the challenge of efficiently managing data mixtures in pretraining by continuously adapting the data distribution based on scaling laws, without requiring proxy models or significant computational overhead. The primary contribution lies in the use of per-domain scaling laws to estimate learning potential and dynamically adjust the data mixture during model training. Experimental results demonstrate that ADO can achieve comparable or better performance than prior methods, while maintaining computational efficiency across various model scales.

**Strengths:**

1.	ADO utilizes online scaling laws to dynamically adapt data selection during training, replacing the need for pre-trained proxy models or multi-staged processes. It requires no prior domain knowledge or external models, making it highly practical and versatile.
2.	The paper presents empirical evaluations demonstrating ADO's effectiveness across multiple benchmarks and datasets. ADO outperforms or matches existing baselines on several downstream tasks while incurring a small computational overhead (e.g., less than 0.4% additional wall-clock time).
3.	ADO offers a practical, scalable solution for data selection in large model training, with the potential to significantly reduce computational waste. The method aligns with the growing emphasis on efficient resource utilization in AI research, especially as models scale in size and cost.

**Weaknesses:**

1.	Figure 4 reveals that fewer than half of the datasets achieve minimal perplexity under the ADO algorithm, indicating potential limitations in the algorithm’s generalization capability across diverse scenarios. The paper would benefit from a more detailed analysis of ADO's applicability.
2.	The algorithm employs heuristic choices for key parameters, such as the exponential moving average in credit assignment and smoothing coefficients, which may impact the robustness and consistency of ADO across different datasets and tasks due to the lack of a more systematic basis for parameter selection.
3.	Symbols such as $\gamma_2$ appears in key formulas but might be clarified further with more detailed context, especially for readers unfamiliar with its role in the algorithm.
4.	Although ADO highlights low computational overhead as a strength, the paper lacks a systematic computational complexity analysis and comparative experiments. The absence of quantitative and visual evidence of its efficiency limits the clarity of its computational advantages.

**Questions:**

1.	In ADO's data selection process, some datasets might receive minimal training due to low selection probabilities, potentially leading to under-representation. Have any additional measures been implemented to balance data distribution and ensure adequate representation across datasets?
2.	How sensitive is ADO to changes in its hyperparameters, such as the smoothing coefficient (γ1) and update intervals? Would different tasks or datasets require significant parameter tuning?
3.	Given the current limitation in modeling solely intra-domain interactions, have the authors considered potential methodologies to incorporate inter-domain interactions within the ADO framework? What approaches might be feasible for capturing these cross-domain dynamics to enhance the model’s adaptability and performance in tasks where inter-domain relationships play a significant role?
4.	ADO is designed to be task-agnostic, but certain applications may benefit from targeted data selection. Could ADO be extended to incorporate task-specific objectives without compromising its efficiency?

---

> ### Author Response · Authors · 2024-11-19
> **Response to Reviewer 7Mvd (1/2)**
>
> We thank the reviewer for their thoughtful review and insightful questions.
>
> **W1**: To clarify, the right half of Figure 4 shows perplexity on training domains, all within The Pile (the training dataset). The observed phenomenon--that ADO does not typically achieve the lowest perplexity on training domains--is interesting but does not indicate a limitation in generalization capability. In fact, ADO achieves the lowest perplexity of all methods on the two datasets that are **not** used in training: SlimPajama and FineWeb, and both are large, diverse, and high-quality text datasets that are considered to be better than The Pile. Hence, Figure 4 in some sense shows that ADO actually has superior generalization compared to other methods. Of course, there is a much more philosophical and open question of what generalization really means for LLMs.
>
> **W2**: Both $\gamma_1$ and $\gamma_2$ are akin to the momentum coefficients in the optimization literature [Defazio et al, NeurIPS2024] and 0.9 is a fairly common value.
> We did not actually tune these values but we believe they should be fairly robust. The smoothing parameter s has a larger effect on the behavior of the algorithm so we have conducted an ablation on the effect of s at the 125M scales in the appendix (Figure 9). Please see our overall response for more details..
>
> **W3**: $\gamma_1$ controls the strength of decay for the exponential moving average on the credit assignment, which gives history $h$. This is analogous to the $\beta_1,\beta_2$ parameters in the Adam optimizer. $\gamma_2$ controls the linear combination between the temporal averaged policy and the current preference distribution--a larger value of $\gamma_2$ makes the policy more adaptive and less stationary. This is similar to Defazio et al.  We apologize for any confusion here and have clarified the text accordingly.
>
> **W4**: This is an interesting point of comparison. Consider the comparison with methods that use proxy models. Empirically, training a 124M model (as a proxy for 1.3B) takes over **10 hours** excluding the post processing time. On the other hand, ADO adds less than 20 minutes to training time, which it uses to fit scaling laws. Note also that the cost of fitting scaling laws is independent of model size.
>
> Theoretically, the computation cost of ADO depends on the rate of convergence of L-BFGS, which can be problem-dependent but is generally efficient. The complexity of fitting per-domain scaling laws scales with the number of domains and the granularity of the initialization search grid. Suppose $G$ is the granularity of discretization for a parameter (e.g., 7), and $K$ is the number of domains, the complexity of fitting would be $O(G^3 K)$, which grows linearly with the number of domains. However, we can easily parallelize multiple L-BFGS runs on TPUs (or GPUs), which is why ADO does not add much time in practice (as we mentioned above, 0.04% training time in a 3 day run which is less than 20 minutes, a number that can likely be optimized further). We have added this analysis to Appendix A.1.
>
> **Q1**: It is a good point and ADO is in fact designed to prevent extreme over- or under- representation. In particular, the power transformation in Eq 2 with smoothing parameter $s<1$ smooths out the credit assignment and leads to a more balanced weighting across domains. We also clip the probabilities to avoid under-representation all together (see Appendix C). The prior distribution $\mu$ also offers another way to assign more emphasis on domains that are considered important.
>
> **Q2**: While training on another dataset of the same size as The Pile would require significant additional time and compute resources, we note that we did not tune any hyperparameters differently between the 124M and the 1.3B which shows robustness across model scale. Of course, like all hyperparameters, the optimal values could be different for different datasets. Also, we added a sweep over values of the smoothing parameter $s$ in Table 3 of the Appendix and found that multiple values of $s$ work well. Even not using any smoothing at all ($s=1$) outperforms the baselines, though having some smoothing is even better.
>
> **Q3**: This is a good question and an important direction for future investigation. One approach could be to use scaling laws that model interactions between domains, for example by introducing parameters to the scaling laws that account for pairwise interaction. The challenge of such an approach is the increase in parameters could make fitting the scaling laws more challenging or impossible. In our preliminary exploration, we found the signal-to-noise ratio for this estimation to be too high to get a reliable estimate.

---

> > ### Author Response · Authors · 2024-11-19
> > **Response to Reviewer 7Mvd (2/2)**
> >
> > **Q4**: This is a very good question. The current version of ADO does not immediately extend to targeted data selection but the idea of using scaling laws could still be useful. For example, one could design a scaling law for the target dataset and go from there. However, it would be hard to do this without sacrificing performance since there is no way to know the performance of the model on the target data at any time during training without actually evaluating them.
> >
> > **Reference**
> >
> > [2] The Road Less Scheduled. Defazio et al, NeurIPS 2024.

---

> > > ### Author Response · Authors · 2024-11-24
> > > **Follow-up**
> > >
> > > Thank you again for the thoughtful reviews and valuable feedback. We would appreciate it if you could let us know if our responses and revision have addressed your concerns and whether you still have any other questions about our work.
> > >
> > > We would be happy to continue the discussion or address any additional comments.

---

> ### Comment · Reviewer_7Mvd · 2024-11-26
> **Response**
>
> Thanks for authors' detailed responses. I think that they have adressed my concerns. I will remain my positive score. Good Luck!

---

### Official Review · Reviewer_monZ · 2024-11-03

**Soundness:** 2
**Presentation:** 2
**Contribution:** 2
**Rating:** 5
**Confidence:** 3

**Summary:**

This paper proposes a novel method ADO to adaptively sample data from different domains based on their contribution to model training, which is estimated from scaling law in an online manner. Empirical results on different data sets demonstrate the effectiveness of the proposed method.

**Strengths:**

The proposed method is introduced clearly and easy to understand

**Weaknesses:**

- Several parts of ADO lack sufficient motivation
- Empirical results seem not supportive enough to fully evaluate the proposed method

**Questions:**

- The smoothing parameter $s$ for computing the credit assignment scores seems a bit confusing. By using $s<1$, I suppose $\lambda_k$ indeed changes drastically when $h_k$ is close to 0? Should using $s<1$ encourages $\lambda_k$ to be larger than $h_k$ especially when $h_k$ is originally close to 0? Some further discussion on the effect of this hyper-parameter may be necessary.
- Also, can we consider other types of functions to compute $\lambda_k$ from $h_k$? Some empirical comparison on different types of functions may help others better understand how the credit assignment score affects model training, and why the authors reject simply using $\lambda_k=h_k$ here.
- Experiments are limited to medium scale models (124M and 1.3B), and the performance gain seems to decrease with increasing model size. Such a tendency can naturally make one wonder if the performance gain can be smaller or even diminish on larger models (e.g., 2.3B or 7/8B)? Some more empirical results may be necessary to better support the proposed method.
- While the proposed method includes many hyper-parameters (e.g., the smoothing parameter $s$ mentioned above), the authors did not provide sufficient analysis on how these hyper-parameters can affect the final performance. These results should be necessary to demonstrate that ADO can be robust to different values of these hyper-parameters.
- Despite showing the final sampling probabilities in Figure 5, the authors may also consider further showing how $\lambda_k(t)$ and $\frac{\partial}{\partial n} \hat{L}_k(n)$ changes with training iteration $t$. These results may help others better understand how these two terms affect the sampling probabilities of different domains and how they can affect model training.

---

> ### Author Response · Authors · 2024-11-19
> **Response to Reviewer monZ**
>
> We thank the reviewer for their thoughtful review and questions. We hope our response below addresses your concerns.
>
> **W1**: We apologize for the lack of motivation in the paper. We have added more motivation in the text, but could you please elaborate on which parts you think lack motivation so we could better address them?
>
> **W2**: Could you elaborate on which results do not support the proposed method and claims we made? The evaluations in Table 1 represent a strong improvement for ADO in comparison with prior literature. For example, ADO improves over the balanced baseline by 3.3% (1.3B) and 1.3% (124M) on average over benchmarked tasks. In Table 2 of [Fan et al, ICML 2024], both DOGE and DoReMi [Xie et al, NeurIPS 2023] improve over the balanced baseline by 1.7% and 0.9%, respectively. We note the benchmarked tasks between these two papers are not exactly the same, but this gives a sense of the scale of improvement. More importantly, ADO is **completely online** whereas almost all prior methods require proxy models and additional computation. We believe this is a major appeal of ADO.
>
> **Q1**: We are not sure if we understand the question. Regardless of the value of h, this transformation always smooths it. Concretely, higher probability domain will have lower probability and lower probability domain will receive higher probability. In the extreme for $s=0$, the $\lambda_k(t) = h_k(t)^0 = 1$ for all $k$, so the it would be uniform across all domains.
> The motivation is similar to having a higher temperature for a softmax distribution, but in our preliminary exploration we found power transformation worked better. The reason why we choose to smooth the distribution is because our estimation is unlikely to be perfect so we want to avoid assigning either very high probability or very low probability to the domains.
>
> **Q2**: This is a great question. There are certainly many choices one can make about this function. Following your previous question, we have conducted an ablation study on the effect of s in Figure 9 of the appendix. We can see that larger s are usually better. This is reasonable since smaller $s$ leads to less credit assignment. On the other hand, $s=1$ which would make $\lambda_k(t) =  h_k(t)$ leads to slightly worse performance. This aligns with our intuition that some degree of smoothing is beneficial since using history to do credit assignment is not perfect.
>
> **Q3**: We fully agree that it would be interesting to test ADO on larger models. Unfortunately, pretraining a 8B model is out of reach given the resources available to us, and arguably, most academics. For reference, our 1.3B model takes 3 days to train. An 8B model would not only take more time per step but also requires more data to train. We believe that it could take much more than 8 times as long to train a single 8B model with the resource we have (i.e., > 24 days). On the other hand, we believe that a 1.3B model is well above the standard practice for data selection methods at an academic conference. For example, DoGE [Fan et al, ICML 2024] ran experiments on models up to 684M.
>
> **Q4**: Thank you for this comment. Both $\gamma_1$ and $\gamma_2$ are akin to the momentum coefficients in the optimization literature [Defazio et al, NeurIPS 2024] and 0.9 is a fairly common value for momentum. We did not actually tune these values at all, but we believe they should be fairly robust. As you have pointed out, $s$ has a larger effect on the behavior of the algorithm so we have conducted a new ablation on the effect of $s$ at the 125M scales in Table 3 of appendix. Please see our overall response for more details.
>
> **Q5**: Thank you for the suggestion. We have included these plots for $\lambda_k(t)$ (Figure 10) and $\frac{\partial}{\partial n} \hat{L}_k(n)$ (Figure 11) in the appendix.
>
> We hope our response, revision and new ablation address your concerns. We'd also be happy to continue the discussion.
>
> **Reference**
>
> [1] DoGE: Domain Reweighting with Generalization Estimation. Fan et al, ICML 2024.
>
> [2] DoReMi: Optimizing Data Mixtures Speeds Up Language Model Pretraining. Xie et al, NeurIPS 2023.
>
> [3] The Road Less Scheduled. Defazio et al, NeurIPS 2024.

---

> > ### Author Response · Authors · 2024-11-24
> > **Follow-up**
> >
> > Thank you again for the thoughtful reviews and valuable feedback. We would appreciate it if you could let us know if our responses and revision have addressed your concerns and whether you still have any other questions about our work.
> >
> > We would be happy to continue the discussion or address any additional comments.

---

> > ### Comment · Reviewer_monZ · 2024-11-26
> >
> > Thank you for your detailed responses that have clarified some of my previous concerns. Here are some of my further comments on your work:
> > - The explanation on how we set the value of $s$ seems a bit confusing. By comparing Figure 5 on the sampling distribution (which corresponds to $\pi(t)$ that is also related to $h(t)$ and $\rho(t)$) with the additional Figure 10 on $\lambda(t)$ during the training process, it seems that $\lambda(t) \propto (h(t))^s$ should actually make the difference between different distribution **larger**. You can also consider the plot of function $y=x^{1/2}$ and $y=x$ with $x \in [0,0.5]$ following the range of $\pi(t)$ in Figure 5. Indeed, I suppose there should be some optimal value of $s$ as is also indicated by the new Table 3, and please correct the above argument for me if you find there is anything wrong.
> > - After checking the comments from other reviewers, I note that reviewer 26JP shares similar opinions to me that empirical improvements on smaller models may not be generalized to larger models, especially considering the fact that the improvements from 124M model to 1.3B model already becomes smaller (just comparing the second best "Natural" to your method). This may be a critical weakness for your work that deserves some discussion, if the available computational resources just forbids you to complete more experiments with larger scales.

---

> > > ### Author Response · Authors · 2024-11-27
> > >
> > > Thank you for getting back to us. We are glad that we are able to resolve part of your concerns.
> > >
> > > Regarding the first point, from a technical perspective $\lambda \propto h^s$ simply makes $\lambda$ smoother (higher entropy) than $h$. We give a concrete example below. The goal is to  avoid “over credit assignment” where domains that are highly represented in recent history do not dominate the credit assignment because history may not be the perfect indicator of credit assignment.
> > >
> > > As shown in Table 3, some degree of credit assignment is desirable for ADO’s performance which confirms the benefit of credit assignment, but non-smoothed credit assignment ($s=1$) can hurt performance which corroborate our hypothesis that using the history directly does not lead to the best credit assignment. Like all hyperparameters in machine learning, the absolute best value can be problem-dependent, which is not necessarily unique to our work. We hope that this answers your question but if we did not understand you correctly, please let us know.
> > >
> > > Regarding the second point, we agree that this warrants more detailed discussion and have included a more detailed discussion of the results in Appendix D.1 (referred to in the main text). The high level takeaway is that while there are tasks that do not improve as much at 1.3B scale, there is also a task (LAMBADA) where ADO does worse at 124M scale but better at 1.3B scale. The variability in a wide collection of individual tasks is not uncommon for LLMs. In terms of average performance, the relative improvement (improvement normalized by the error) for 124M and 1.3B scales are 1.3% and 1.2% respectively so its average improvement at different scales is actually consistent.
> > >
> > > Regarding larger models, we agree that we do not know for sure whether ADO would work with bigger models and we already include this as a limitation section of the paper but would be happy to revise it further if you have any suggestions.
> > >
> > > ------
> > >
> > > Example of how $\lambda \propto h^s$ makes the distribution smoother (higher entropy):
> > > $$h = [0.1, 0.9], s = 0.5$$
> > > $$h^s = [0.316, 0.948]$$
> > > $$\lambda = h^s / \sum_{i=1}^2h^s(i) \approx [.25, 0.75]$$
> > >
> > > so clearly $\lambda$ is higher entropy (i.e., smoother) than the original $h$, with $s=0.5$.

---

### Official Review · Reviewer_26JP · 2024-11-04

**Soundness:** 3
**Presentation:** 3
**Contribution:** 2
**Rating:** 5
**Confidence:** 3

**Summary:**

The authors propose ADO, an online data selection method that adjusts data distribution dynamically across various domains by leveraging domain scaling laws. Without requiring a proxy model or external knowledge, ADO forecasts the model's loss on different data domains and automatically modifies the training data distribution according to each domain's learning potential. The experiments indicate that ADO surpasses baseline methods overall, with only a minimal increase in clock time.

**Strengths:**

1. The motivations presented in Section 2 highlight significant concerns, and the proposed method properly addresses these issues.
2. ADO outperforms the baselines overall, offering online data selection without the requirement of external knowledge or proxy models.
3. The authors present several interesting observations that align with our intuition and the literature (e.g. line 428, 409, 431).

**Weaknesses:**

1. Although the 1.3B model is not small, ADO is especially practical and relevant for LLMs with at least 8B parameters—the size typical of many popular "small" LLMs such as Llama 3.1 8B. The paper's relevance and importance might have been enhanced if the authors had conducted experiments with language models of at least 8B parameters
2. The performance improvement achieved by ADO does not seem significant enough, especially on the 1.3B model. More problematically, Table 1 shows diminishing returns as the model scale increases from 124M to 1.3B parameters. For example, 1.3B-ADO only outperforms 4 out of 7 downstream tasks, whereas 124M-ADO outperforms 6 out of 7 tasks. Additionally, the gap between the average score of ADO and the second-best baseline is less pronounced in the 1.3B model. This suggests that ADO may not scale well as the number of parameters increases.
3. Although the authors propose interesting future research directions (e.g. Section 6), incorporating some of these suggestions—such as learning rate scheduling—into the current paper would make it more complete and thorough.

**Questions:**

1. Figures 6, 7, and 8 show the training loss for ADO only. I'd like to see how quickly ADO's training loss decreases compared to the baselines. I assume that ADO converges faster, but it would be interesting to observe how it performs relative to the baselines.

2. Can ADO be effectively applied to vision data and tasks? It would be of great importance if ADO could be used in training vision models that require multiple domains, such as large vision-language models (LVLMs) and diffusion models for image generation.

---

> ### Author Response · Authors · 2024-11-19
> **Response to Reviewer 26JP**
>
> We thank the reviewer for their careful analysis and thoughtful feedback. We hope our response below addresses your concerns.
>
> **W1**: We agree that it would be interesting to test ADO on larger models. Unfortunately, pretraining a 8B model is out of reach given the resources available to us, and arguably, most academics. For reference, our 1.3B model takes 3 days to train. An 8B model would not only take more time per step but also requires more data to train [Hoffman et al, 2022]. We believe that it could take more than 8 times as long to train a 8B model with the resource we have. On the other hand, we believe that a 1.3B model is well above the standard practice for data selection methods at an academic conference. For example, DoGE [Fan et al, ICML 2024] ran experiments on models up to 684M.
>
> **W2**: While ADO’s improvement over the "*Natural*" baseline is relatively modest (0.7% at 124M and 0.5% at 1.3B), as we discuss in the paper, the “*natural*” baseline turns out to be a strong baseline that was not commonly tested in prior work.
>
> When we compare against the “*balanced*” (i.e., each domain receives the same weight) baseline, as does most prior work in this area, the magnitude of improvement for ADO is significant in comparison with existing literature. Per Table 2 in [Fan et al, ICML 2024], both DOGE and DoReMi [Xie et al, NeurIPS 2023] improve over the balanced baseline by 1.7% and 0.9%, respectively. Meanwhile, ADO improves over the balanced baseline by 3.3% (1.3B) and 1.3% (124M). While the benchmarked tasks differ somewhat and the settings are different (5-shot vs 0-shot), this shows that the scale of improvement is in line with or larger than prior work. In fact, ADO’s improvement relative to the “balanced” baseline **increases** with scale.
>
> In addition, by the nature of scaling law, most, if not all, performance metrics would become more difficult to improve with larger scales because the room to improve becomes smaller, so it is unrealistic to expect that absolute performance gain would stay the same at different scales. Concretely, the error for “natural” at 124M is $1-0.463 = 0.537$ and ADO improved over “natural” by $0.07$ so the relative improvement is $0.007/0.537=0.013$. The error for “natural” at 1.3B is $1-0.585 = 0.415$ and ADO improved over “natural” by $0.05$ so the relative improvement is $0.005/0.415=0.012$. Based on this calculation, we can see that the relative improvement of ADO at different scales is consistent.
>
> **W3**:  The effect of learning rate schedule on the learning curve’s functional form is a very new research direction. In fact, the first paper on the topic (that we are aware of) [Tissue et al, 2024] only came out very recently after we had finished our experiments. There are many things that we don’t fully understand about these methods, and we do not want to introduce too many moving components to our method because these methods alone would be worth one or more stand alone papers. As such, we believe these directions are better left for future works.
>
> **Q1**: Good question--we have added this information in Figure 9 of the appendix. An interesting observation is that ADO’s training loss actually does not decrease faster than baselines because ADO prioritizes the domains with higher learning potential, which does not necessarily mean low loss. In fact, we see that ADO tends to select higher quality data which naturally have higher entropy which would lead to higher training loss.
>
> **Q2**: We believe that it should be possible to apply ADO or ADO-like methods to other domains as long as the loss curve looks like a well-behaved power law. Without other interpretations, ADO can be seen as a method that prioritizes data points whose loss is going down faster than the other ones. However, the information-theoretic interpretation might not apply depending on the exact training objective. It would be very interesting to explore this in future works.
>
> **Reference**
>
> [1] DoGE: Domain Reweighting with Generalization Estimation. Fan et al, ICML 2024.
>
> [2] DoReMi: Optimizing Data Mixtures Speeds Up Language Model Pretraining. Xie et al, NeurIPS 2023.
>
> [3] Scaling Law with Learning Rate Annealing. Tissue et al, 2024.
>
> [4] Training Compute-Optimal Large Language Models. Hoffman et al, 2022.

---

> > ### Author Response · Authors · 2024-11-24
> > **Follow-up**
> >
> > Thank you again for the thoughtful reviews and valuable feedback. We would appreciate it if you could let us know if our responses and revision have addressed your concerns and whether you still have any other questions about our work.
> >
> > We would be happy to continue the discussion or address any additional comments.

---

> ### Comment · Reviewer_26JP · 2024-11-25
>
> Thank you for addressing the concerns raised in my review. I would like to elaborate on these points.
>
> On Experiment Scale (W1)
> - I understand that training an 8B model may be infeasible given your resource constraints. However, I must respectfully disagree with the justification that a 1.3B model is sufficient for relevance. While it is true that academic research often operates under resource limitations, the field of LLMs is evolving rapidly, and what was considered “state-of-the-art” or “adequate” a year ago may no longer hold true. For example, DoGE's experiments, very likely completed in 2023, represent a landscape that has shifted quite significantly by now, nearing 2025. Models like Llama have become widely used benchmarks and hold substantial weight in the research community.
>
> - To make a lasting contribution, I believe your method should demonstrate its practicality and relevance to the contemporary scale and settings where it could realistically be applied. This would elevate the significance of your work beyond being an interesting but narrowly scoped academic exercise.
>
> Magnitude of Improvement and Task Performance (W2)
> - Your clarification on the relative improvement over the “natural” and “balanced” baselines is appreciated. However, I remain unconvinced by your argument that a modest improvement of 0.5-0.7% over “natural” is consistent. The weakness in ADO’s performance at larger scales is not solely due to diminished relative improvements. I noted that while ADO with the 124M model outperforms baselines on 6 out of 7 downstream tasks, this drops to 4 out of 7 tasks with the 1.3B model. This raises concerns about ADO’s consistency and robustness as model scale increases. This trend is particularly important since it would be problematic if ADO’s benefits do not consistently increase with scale relative to the “balanced” baseline. However, the task-level performance metrics raise concerns and further analysis or discussion of this discrepancy would be valuable.
>
> While I still see value in sharing this work with the community, I stand by my assessment that the current contribution remains marginally above the acceptance threshold. The primary concerns—namely, the scale of experiments, the practical relevance of ADO, and the inconsistency in task-level performance at larger scales—prevent the paper from being a strong acceptance. To elevate this work beyond “barely acceptable,” addressing these concerns in meaningful ways (e.g., experiments with more contemporary model architectures, stronger demonstrations of generalization, and a discussion on task-level performance trends) would be necessary.

---

> > ### Author Response · Authors · 2024-11-26
> >
> > Thank you for getting back to us. We are glad that we are able to address some of your concerns.
> >
> > We can sympathize with the concern about testing at large scales, but we also believe that the most important factor of open-ended research is original conceptual contributions that open up new directions. We've demonstrated that ADO is much more scalable (i.e., compute-efficient) than prior works and consistently outperforms prior works across different scales, even if not at the absolute frontier. Since it is beyond our resource constraints to train an 8B model, we will focus on clarifying any potential misconceptions or questions.
> >
> > > While it is true that academic research often operates under resource limitations, the field of LLMs is evolving rapidly, and what was considered “state-of-the-art” or “adequate” a year ago may no longer hold true. For example, DoGE's experiments, very likely completed in 2023, represent a landscape that has shifted quite significantly by now, nearing 2025. Models like Llama have become widely used benchmarks and hold substantial weight in the research community.
> >
> > > experiments with more contemporary model architectures [...]
> >
> > Llama2 was released in 2023 (i.e., before DoGE) and was widely popular, so we do not see how the popularity of Llama2 or Llama3 affects the merits of DoGE or our work. In fact, **the model architectures of Llama2 and Llama3 are not significantly different**. Both use a similar decoder-only architecture, but Llama3 features a longer context window (which is not directly related to our work).
> >
> > The main difference in pretraining between them lies in the composition and filtering of the training data. Llama3 was trained on larger datasets that include multilingual and code data, filtered with more sophisticated techniques. The other parts of the pretraining pipeline have not changed significantly. This actually underscores the importance of research into automated data selection for pretraining, as it directly complements such advancements. We also want to clarify that we have not claimed to be “state-of-the-art,” as such a description is inherently difficult to quantify for foundation models.
> >
> > > To make a lasting contribution, I believe your method should demonstrate its practicality and relevance to the contemporary scale and settings where it could realistically be applied. This would elevate the significance of your work beyond being an interesting but narrowly scoped academic exercise.
> >
> > We emphasize that ADO is more practical than the existing methods for data selection because it is completely online and incurs almost no additional cost. Furthermore, not only is it more practical, but it also outperforms them by a large margin so it is relevant in this line of literature.
> >
> > > This trend is particularly important since it would be problematic if ADO’s benefits do not consistently increase with scale relative to the “balanced” baseline. However, the task-level performance metrics seem to contradict this claim, and further analysis or discussion of this discrepancy would be valuable.
> >
> > To clarify, ADO outperforms the “balanced” baseline (i.e., the baseline used in prior work) in **every downstream task at both scales**. In fact, the improvement over balanced is **larger** as the model becomes larger: the absolute average improves by 1.3% at 124M and 3.3% at 1.3B.  (note that both “natural” and ADO are original contributions of this work that do not exist in prior work). We have also added a detailed discussion on per-task performance in appendix D.1.
> >
> > In addition to the downstream evals, we also evaluate perplexity on diverse high quality datasets like SlimPajama and FineWeb. The results demonstrate that our method consistently outperforms all baselines. Given that high-quality datasets are widely regarded as essential for building better foundation models, ADO’s ability to excel in these settings is highly desirable.
> >
> >
> > > stronger demonstrations of generalization [...]
> >
> > We would appreciate clarification on what a stronger demonstration of generalization might be, aside from simply scaling the model size. We have demonstrated that ADO trained with the Pile performs better than other methods on different high-quality datasets like SlimPajama and FineWeb. We believe that this actually demonstrates a desirable generalization ability of ADO.

---

> ### Comment · Reviewer_26JP · 2024-11-27
>
> I have additional concerns after further reflection, particularly regarding comparisons to prior work and the lack of theoretical contributions in this paper. I outline these below.
>
> On Comparisons to DoReMi and DoGE
> - It is surprising that the rebuttal does not discuss DoReMi (NeurIPS 2024) when addressing conventional model sizes for academic conferences. DoReMi explicitly targets the training of 8B models, as stated clearly in their abstract, and evaluates their method at the 8B scale, showcasing their relevance in the current landscape.
>
> - Furthermore, both DoReMi and DoGE provide significant theoretical contributions. For instance, DoReMi includes a rigorous theoretical contribution (Appendix D), while DoGE complements its experimental results with theoretical justification (Appendix B). These contributions enhance the broader impact and rigor of their work, even when experiments are limited to smaller scales (DoGE). Experiments in DoReMI are more comprehensive, testing 8B models on more datasets than this work does.
>
> - In contrast, ADO neither performs experiments at larger scales nor offers meaningful theoretical contributions. While the empirical results are interesting, the absence of theoretical justification or deeper analysis significantly weakens the paper, particularly when compared to these prior works. This is especially problematic for a method that claims to introduce an original approach but does not delve into why ADO (or some of its elements) works at a fundamental level.
>
> On Downstream Task Performance and the “Natural” Baseline
> - While the authors argue that both “natural” and ADO are original contributions of this work, the focus of this paper is on ADO. As such, ADO must consistently outperform the “natural” baseline to demonstrate its value. For the 1.3B model, ADO only outperforms “natural” on 4 out of 7 tasks, compared to 6 out of 7 tasks at 124M. In particular, the “natural” baseline outperforms ADO on ARC-E, SciQ, and LogiQA2.
>
> Empirical Nature of the Paper
> - The paper is heavily empirical and lacks theoretical justification for why ADO works. While simplicity and practicality are valuable attributes, they are not sufficient to compensate for the absence of theoretical insights or deeper analysis. In this regard, the paper falls short of the standards set by prior works such as DoReMi and DoGE, both of which balance empirical results with robust theoretical contributions.
>
> After reevaluating the paper, I have decided to lower my score to 5 (“below acceptance threshold”). While ADO demonstrates merit in terms of compute efficiency and modest empirical improvements, the paper falls short in its theoretical and experimental rigor relative to existing literature. Specifically:
> - ADO’s inability to consistently outperform the “natural” baseline at larger scales undermines its claim of scalability.
> - The lack of theoretical contributions or deeper analysis makes the paper overly reliant on its empirical results, which are not sufficiently comprehensive.
> - Comparisons to prior work reveal significant gaps in scope, relevance, and contribution.
>
> While ADO has potential, the paper’s current form does not meet the standard of significant contribution required for acceptance. I hope the authors will consider addressing these concerns in future revisions, as there is a foundation here for meaningful work if the necessary rigor and scope (as shown in DoReMI) are added.

---

> ### Author Response · Authors · 2024-11-27
>
> Thank you for the additional comments and continued engagement. We will try to address these new comments below.
>
> DoReMi did indeed contain experiments using far more resources than most academics would have access to since it is from one of the largest industry labs.
> We already acknowledge in the limitation of the original draft that we do not know how ADO will perform on larger models, and we have revised the limitation section to explicitly mention the 8B model of DoReMi and would be happy to revise this section more if you have further suggestions.
>
> > The lack of theoretical contributions or deeper analysis makes the paper overly reliant on its empirical results, which are not sufficiently comprehensive.
>
> > [...] prior works such as DoReMi and DoGE, both of which balance empirical results with robust theoretical contributions.
>
> Our paper is indeed empirical in nature and we do not claim any theoretical contribution. However, we believe that the existing literature in this area is also heavily empirical in nature. The two prior works you mention here (DoReMi and DoGE) provide interesting derivations and analyses, but these do not necessarily amount to “theoretical justification” for why their methods work.
>
> For example, in DoGE (Appendix B), the authors derive the DoGE update rule as an optimization problem--but it is not a proof for why DoGE works. In fact, from DoGE (Appendix C.5):
>
> *“the performance of the proxy model falls behind the base model with resampled dataset with DOGE domain sampling weights. It is even worse than the baseline with uniform domain weights.“*
>
> If the derivation actually explained DoGE’s efficacy, then the proxy model should do better than uniform weights because it is actually trained with the derived update rule. The fact that the proxy model does *worse* indicates that the motivation does not match what the algorithm is actually doing. As such, it stands to reason that DoGE’s success is also empirical. In terms of empirical results’s comprehensiveness, we believe that the scope of our results is comparable to DoGE and we also conduct experiments on larger models.
>
> In DoReMi (Appendix D), the authors constructed a toy data generating process for which there exists a set of weights that is better than uniform, but this is not a proof that DoReMi would actually find these weights. The fact that DoReMi works on this toy model was also an empirical observation.
>
> > ADO’s inability to consistently outperform the “natural” baseline at larger scales undermines its claim of scalability.
>
> > While the authors argue that both “natural” and ADO are original contributions of this work, the focus of this paper is on ADO. As such, ADO must consistently outperform the “natural” baseline to demonstrate its value. For the 1.3B model, ADO only outperforms “natural” on 4 out of 7 tasks, compared to 6 out of 7 tasks at 124M. In particular, the “natural” baseline outperforms ADO on ARC-E, SciQ, and LogiQA2.
>
> In our opinion, the contribution of a work should be measured against the existing literature, and we have demonstrated that ADO outperforms the existing techniques and popular baselines in the literature while being much more efficient. Furthermore, for LAMBADA, natural outperforms ADO in 124M but underperforms in 1.3B, indicating that there can be variability in individual tasks. In terms of the more robust average performance, the relative improvement of ADO over natural is consistent across scales (1.3% and 1.2% respectively). We believe that this supports ADO’s value.
>
> Further, consistent improvement in all downstream tasks over prior works is not the standard of the literature for pretraining to the best of our knowledge. For example, DoGE does not consistently outperform the DoReMi in downstream performance (Table 2 of DoGE). In comparison, ADO outperforms DoReMi in 6/7 tasks and performs the same in 1/7 at both scales.
>
> We hope this clarifies these points and would be happy to continue the discussion.

---

### Author Response · Authors · 2024-11-19
**Summary of revision**

We thank all of the reviewers for their questions and feedback, much of which we have already included in our revised manuscript. Please see the revised manuscript which answers many of the questions from the reviews--changes since the original submission are made in red. We also did some tightening so the text remains within the page limit.

As a quick summary of the changes:

* **Table 3** in the appendix: We conducted an ablation study on sweeping the hyperparameter $s \in [0,1]$ which controls the smoothing during credit assignment. We find that ADO is not particularly sensitive to $s$: of the four values we swept, which span a wide range of possible values (0.1, 0.3, 0.5, and 1.0), all four values resulted in improvements over the “Balanced” baseline and three out of four improved over the “Natural baseline.” Most notably, not using any smoothing at all ($s=1$) still outperforms the baselines, though having some smoothing is better which confirms our hypothesis. (Reviewer monZ, 7Mvd, Axts)

* **Figure 9** in the appendix: Training curves of 1.3B models under each method, including ADO and the baselines (Reviewer 26JP)

* **Figure 10** in the appendix: A plot of the credit assignment function $\lambda_k(t)$ for each domain throughout the course of training. (Reviewer monZ)

* **Figure 11** in the appendix: The estimated speed of learning $\frac{\partial}{\partial n} \hat{L}_k(n)$ on each domain given by the slope of the scaling laws. (Reviewer monZ)

We also edited Section 3 to include more intuition about the hyperparameters and how adjusting them affects ADO.

---

### Meta-Review · Area_Chair_bZ6T · 2024-12-20

**Metareview:**

This paper proposes an algorithm to optimise data distributions accros domains during model training in order to leverage domain-based scaling laws. The claims are supported and validated experimentally. The topic is timely and all reviewers agreed that the results are worth sharing with the community. One remaining question remained unanswered, namely that we do not know for sure whether the proposed method would still work with bigger models than the ones considered in the paper. While this is a valid point, it does not invalidate the results reported in this work. The authors also provided an adequate discussion of this limitation.

**Additional Comments On Reviewer Discussion:**

Reviewers acknowledged the response provided by the authors. Authors provided detailed clarifications addressing the comments of reviewers. One remaining questions remained (see above). However, the most critical reviewer stated that they found the contribution marginally above the acceptance threshold (even though they scored it 5). The general consensus is that this work is solid and insights valuable to the research community.

---

### Decision · Program_Chairs · 2025-01-22

Accept (Poster)